# Sphinganine recruits TLR4 adaptors in macrophages and promotes inflammation in murine models of sepsis and melanoma

Marvin Hering [1,2,3,4,5] ✉, Alaa Madi [1], Roger Sandhoff [6], Sicong Ma[1,7], Jingxia Wu[1,7], Alessa Mieg[1], Karsten Richter[8], Kerstin Mohr[1], Nora Knabe[1,9], Diana Stichling[1], Gernot Poschet [10], Felix Bestvater[11], Larissa Frank [2,12], Jochen Utikal[3,4,5], Viktor Umansky[3,4,5,13] & Guoliang Cui [1,2,7,9,14] ✉

After recognizing its ligand lipopolysaccharide, Toll-like receptor 4 (TLR4) recruits adaptor proteins to the cell membrane, thereby initiating downstream signaling and triggering inflammation. Whether this recruitment of adaptor proteins is dependent solely on protein-protein interactions is unknown. Here, we report that the sphingolipid sphinganine physically interacts with the adaptor proteins MyD88 and TIRAP and promotes MyD88 recruitment in macrophages. Myeloid cell-specific deficiency in *serine palmitoyltransferase long chain base subunit 2*, which encodes the key enzyme catalyzing sphingolipid biosynthesis, decreases the membrane recruitment of MyD88 and inhibits inflammatory responses in in vitro bone marrow-derived macrophage and in vivo sepsis models. In a melanoma mouse model, *serine palmitoyltransferase long chain base subunit 2* deficiency decreases anti-tumor myeloid cell responses and increases tumor growth. Therefore, sphinganine biosynthesis is required for the initiation of TLR4 signal transduction and serves as a checkpoint for macrophage pattern recognition in sepsis and melanoma mouse models.

Toll-like receptor 4 (TLR4) recognizes bacterial lipopolysaccharide (LPS) and subsequently initiates downstream signal transduction, thus driving M1-like macrophage responses. Adaptor proteins, such as myeloid differentiation primary response gene 88 (MyD88) and Toll/interleukin-1 receptor (TIR) domain-containing adaptor protein (TIRAP), are recruited to the intracellular domain of the TLR4 protein. The adaptor proteins in turn recruit signal transduction molecules and eventually activate nuclear factor kappa-light-chain-enhancer of activated B-cells (NF-κB), which translocates into the nucleus, and subsequently induces the expression of genes associated with

[1]T Cell Metabolism Group (D192), German Cancer Research Center (DKFZ), Heidelberg, Germany. [2]Faculty of Biosciences, Heidelberg University, Heidelberg, Germany. [3]Skin Cancer Unit, German Cancer Research Center (DKFZ), Heidelberg, Germany. [4]Department of Dermatology, Venereology and Allergology, University Medical Center Mannheim (UMM), Ruprecht-Karls University of Heidelberg, Mannheim, Germany. [5]DKFZ Hector Cancer Institute at the University Medical Center Mannheim, Mannheim, Germany. [6]Lipid Pathobiochemistry Group (A411), German Cancer Research Center (DKFZ), Heidelberg, Germany. [7]Institute of Health and Medicine, Hefei Comprehensive National Science Center, Hefei 230601, China. [8]Electron Microscopy Core Facility (W230), German Cancer Research Center (DKFZ), Heidelberg, Germany. [9]Helmholtz Institute for Translational Oncology, Mainz (HI-TRON Mainz)—A Helmholtz Institute of the DKFZ, Mainz, Germany. [10]Metabolomics Core Technology Platform, Centre for Organismal Studies (COS), Heidelberg University, Heidelberg, Germany. [11]Light Microscopy Core Facility (W210), German Cancer Research Center (DKFZ), Heidelberg, Germany. [12]Division of Cellular Immunology, German Cancer Research Center (DKFZ), Heidelberg, Germany. [13]Mannheim Institute for Innate Immunoscience (MI3), Medical Faculty Mannheim, University of Heidelberg, Mannheim, Germany. [14]Key Laboratory of Immune Response and Immunotherapy, Center for Advanced Interdisciplinary Science and Biomedicine of IHM, School of Basic Medical Sciences, Division of Life Sciences and Medicine, University of Science and Technology of China, Hefei, China. ✉e-mail: MarvinHering@web.de; gcui@ustc.edu.cn

inflammatory cytokine production and cell morphological changes[1–3]. A key step in this signaling cascade is the recruitment of adaptor proteins to TLR4. Both TLR4 and adaptor proteins have TIR domains, and the homotypic TIR-TIR association promotes the interactions between adaptor proteins and TLR4. Whether the membrane recruitment of adaptor proteins is dependent solely on protein-protein interactions is unknown.

Multiple metabolites, including fatty acids, sphingolipids and psychoactive xenobiotics, have been shown to initiate or regulate TLR4 signal transduction[4–6]. To systemically study which metabolites influence the recruitment of adaptor proteins to TLR4, we quantified 1019 metabolites in LPS-/IFNγ-treated macrophages. Subsequent non-biased bioinformatics analysis revealed that LPS-/IFNγ-treatment induced enrichment of sphingolipids in macrophages. Sphingolipid de novo biosynthesis begins with the condensation of L-serine and palmitoyl-CoA to 3-keto sphinganine (3-KDS), as catalyzed by serine palmitoyltransferase (SPT), with the catalytically active subunit SPT long chain base subunit 2 (Sptlc2). Subsequently, 3-KDS is converted to sphinganine, ceramides and more complex sphingolipids.

To study whether Sptlc2-mediated synthesis of sphingolipids might regulate TLR4 signal transduction, we used a mouse strain with myeloid cell-specific ablation of *Sptlc2*. Deficiency in *Sptlc2* impaired LPS-induced inflammation both in vitro and in vivo. Moreover, sphinganine physically interacted with and promoted the cell membrane localization of MyD88, and restored LPS-induced inflammation in *Sptlc2*-deficient macrophages. Our findings revealed that sphingolipid biosynthesis is required for the initiation of innate immune responses, and highlights sphinganine as a metabolic checkpoint involved in pattern recognition by macrophages.

## Results

### LPS induces sphingolipid biosynthesis in macrophages

To study macrophages, we used the well-established model of bone marrow-derived macrophages (BMDM)[7]. Bone marrow (BM) cells were differentiated to BMDM with M-CSF for 6 days, and subsequently polarized to M1-like or M2-like macrophage activation states (Fig. 1a). To identify metabolites regulating inflammatory signaling in macrophages, we performed a large, targeted metabolomics screening via the Biocrates MxP® Quant 500 XL kit to determine up to 1019 metabolites in BMDM treated with LPS/IFNγ (M1-like macrophages) (Fig. 1b). BMDM treated with IL-4/IL-13 (M2-like macrophages) and PBS-treated BMDM (M0-like macrophages) served as controls. LPS/IFNγ treatment increased the concentrations of the amino acid citrulline, whereas IL-4/IL-13 treatment increased the levels of the amino acid kynurenine, thus echoing previous results indicating that in M1-like macrophages arginine is metabolized to nitric oxide and citrulline, whereas in M2-like macrophages, kynurenine is generated from tryptophan degradation[8] (Fig. 1b; Supplementary Data 1). M2-like macrophages, which are known to depend on fatty acid oxidation, but not LPS-/IFNγ-treated macrophages, which are dependent on glyco-lysis, showed significantly increased levels of the fatty acid eicosenoic acid (20:1) compared to M0-like macrophages (Fig. 1b; Supplementary Data 1). Although no clear differences in the levels of acylcarnitines or bile acids were observed between untreated and LPS-/IFNγ-treated macrophages, metabolites belonging to various lipid subclasses were elevated in LPS-/IFNγ- and IL-4-/IL-13-treated macrophages, including phospholipids (e.g., phosphatidylethanolamines, phosphatidylcho-lines), glycerophospholipids (e.g., phosphatidylglycerols, phosphati-dylinositols) and triacylglycerols (Fig. 1b; Supplementary Data 1). Treatment with LPS/IFNγ and, to a lesser extent, IL-4/IL-13 resulted in significantly greater levels of sphingolipids (ceramides, glycosylcer-amides, dihydroceramides, sphingomyelins and sphingoid bases) than observed in untreated macrophages (Fig. 1b; Supplementary Data 1). Exemplary, sphingoid base d16:0 was significantly enriched in LPS-/IFNγ-treated macrophages compared with untreated macrophages

but was not significantly enriched in IL-4-/IL-13-treated macrophages (Fig. 1b; Supplementary Data 1). To examine whether the LPS-/IFNγ-induced sphingolipid enrichment resulted from enhanced de novo biosynthesis or from recycling pathways, we mined previously pub-lished mouse RNA sequencing data from Nguyen et al. and found that LPS stimulation increased the mRNA levels of *Sptlc2* in BMDM[9] (Fig. 1c). Sptlc2 catalyzes the rate-limiting step in sphingolipid de novo synthesis, in which L-serine and palmitoyl-coenzyme A (CoA) are condensed, thereby forming 3-KDS (Fig. 1d). In contrast, the RNA levels of several enzymes involved in sphingolipid recycling pathways, such as ceramidases (*Asah1* and *Asah2*), sphingomyelinases (*Smpd1*, *Smpd2*, *Smpd3* and *Smpd4*), sphingosine-1-phosphate phosphatase (*Sgpp1*), glucosylceramidases (*Gba* and *Gba2*) and galactosylcer-amidase (*Galc*) were unchanged or even decreased after LPS exposure[9] (Fig. 1c). On the basis of these findings and previous reports, we hypothesized that LPS treatment would specifically enhance Sptlc2-dependent sphingolipid synthesis[10]. Similarly to the RNA sequencing results, Sptlc2 protein levels and ceramide levels were higher in M1-like macrophages than in M0-like or M2-like macrophages (Fig. 1e; Sup-plementary Fig. 1a, b). Inspired by these findings, we generated mice with myeloid cell-specific deficiency in *Sptlc2* (referred to as *Lyz2*-cre or *Sptlc2*-deficient; these are *Sptlc2*^Flox/Flox^ *Lyz2*-cre-positive; Supple-mentary Fig. 1c; for more information see method section). In order to narrow down to the TLR4 pathway specifically, we then performed western blotting to confirm that LPS alone increased the expression of Sptlc2 protein in wildtype (referred to as WT; *Sptlc2*-sufficient; these are *Sptlc2*^Flox/Flox^ *Lyz2*-cre-negative) BMDM[10] and used *Sptlc2*-deficient BMDM samples as negative controls (Fig. 1f; Supplementary Fig. 1d, e). In line with the in vitro results, intraperitoneal injection of LPS increased Sptlc2 protein levels in intraperitoneal and splenic macro-phages (Fig. 1g). Collectively, these results suggested that LPS enhances the expression of Sptlc2 and induces sphingolipid synthesis in macrophages, thus prompting the question of whether macro-phages require Sptlc2 for LPS-induced signal transduction and cell polarization.

### *Sptlc2* deficiency decreases macrophage cell growth and meta-bolic fitness

During BMDM differentiation, the medium of *Sptlc2*-sufficient macro-phages, particularly M1-like macrophages, appeared more yellow than that of *Sptlc2*-deficient counterparts (Fig. 2a). Subsequently, pH mea-surements confirmed that the medium of *Sptlc2*-sufficient macrophages was more acidic than that of *Sptlc2*-deficient macrophages, and M1-like macrophage medium was the most acidic (Fig. 2a; Supplementary Fig. 2a). Changes in medium pH values can be caused by differences in cell numbers, glycolytic activities or both. *Sptlc2*-sufficient macro-phages showed higher confluency than *Sptlc2*-deficient macrophages after LPS stimulation (Fig. 2b; Supplementary Fig. 2b, c). Scanning electron microscopy analysis revealed that *Sptlc2*-sufficient and *Sptlc2*-deficient macrophages had distinct morphologies: The LPS-polarized *Sptlc2*-sufficient macrophages showed a bumpy cell body in the center, from which filopodia and lamellipodia extended, whereas the *Sptlc2*-deficient macrophages showed a flat cell body and very few filopodia (Fig. 2c; Supplementary Fig. 2d). Supplementation with sphinganine, a metabolite downstream of Sptlc2, restored the morphology of *Sptlc2*-deficient macrophages (Fig. 2c; Supplementary Fig. 2d). To test whether *Sptlc2* deficiency might influence cellular metabolism, we performed Seahorse extracellular flux analysis to assess the mitochondrial respiratory and glycolytic states of M1-like and M2-like *Sptlc2*-deficient or -sufficient BMDM (Fig. 2d; Supplementary Fig. 2e). In agreement with findings from a previous study[11], M1-like macrophages, compared with M2-like macrophages, showed higher glycolytic rates at the expense of oxidative phosphorylation (Fig. 2d; Supplementary Fig. 2e). Moreover, *Sptlc2* deficiency decreased the basal and maximal oxygen consump-tion rate (OCR), and the spare respiratory capacity (SRC; difference in

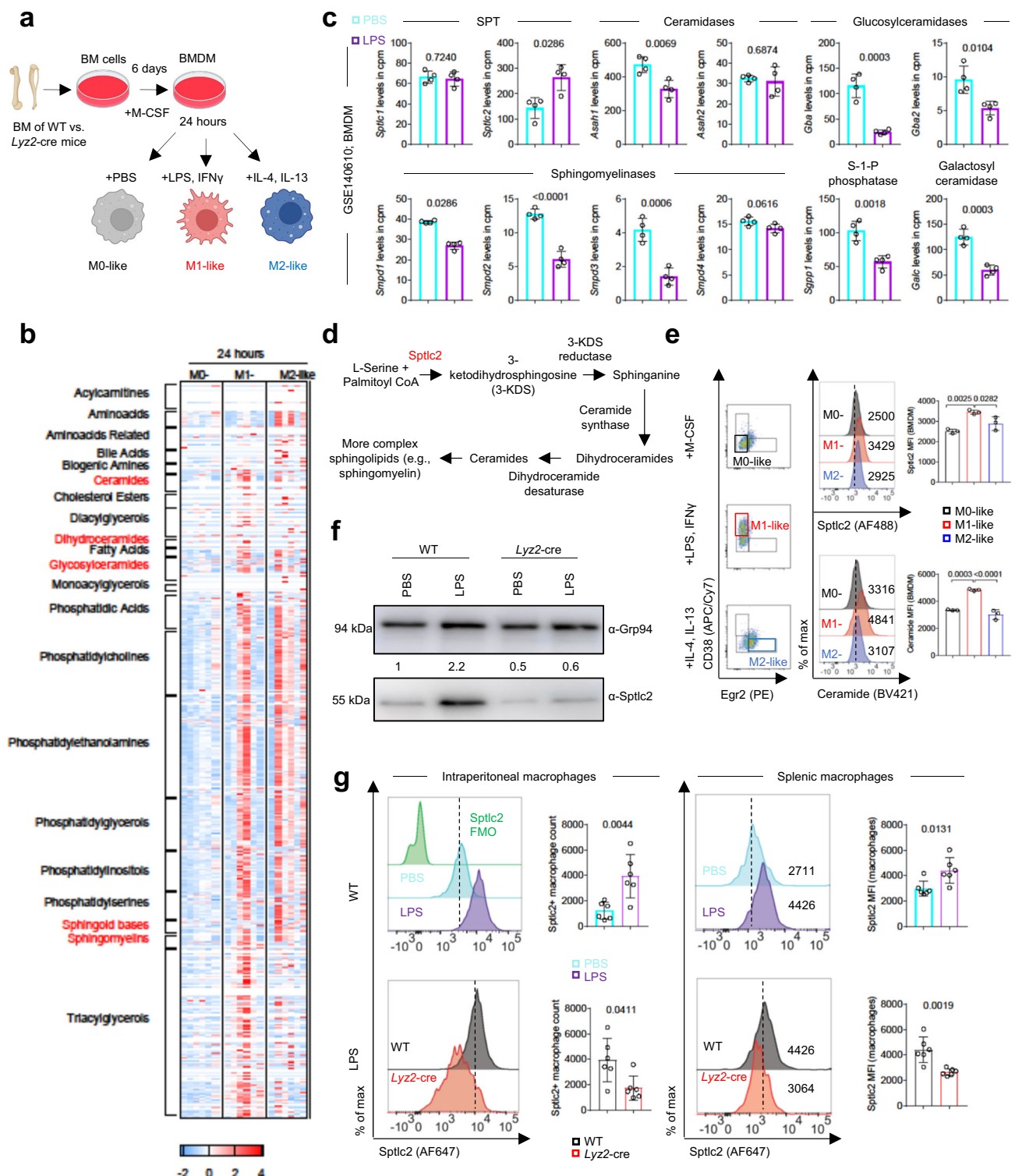

basal and maximal OCR), of M2-like BMDM, as well as the basal extracellular acidification rate (ECAR), glycolytic capacity (maximal ECAR) and glycolytic reserve (difference in glycolytic response ECAR and maximal ECAR) of M1-like BMDM (Fig. 2d; Supplementary Fig. 2e). Collectively, these results suggested that Sptlc2 is required for the LPS-induced cell growth, morphological changes, and metabolic fitness of M1-like macrophages.

To examine the sphingolipid profiles and synthesis in *Sptlc2*-sufficient or *Sptlc2*-deficient macrophages, we cultured *Sptlc2*-sufficient or *Sptlc2*-deficient macrophages with $^{13}C_3{}^{15}N_1$ L-serine for 2 h before

measuring the incorporation of the tracers into sphingolipids (Fig. 2e). Almost all measured endogenous and stable isotope-labeled sphingolipids were significantly diminished in *Lyz2*-cre BMDM, thus suggesting that *Sptlc2* deletion strongly alters the sphingolipidome in macrophages (Fig. 2f; Supplementary Fig. 2f, g). LPS-activated macrophages are characterized by increased cell size compared to untreated macrophages[12]. In agreement with the confluency assay results (Fig. 2b; Supplementary Fig. 2b, c), the percentage of large (FSC-A^high, SSC-A^high) BMDM among the total macrophages was significantly reduced upon *Sptlc2* deletion after LPS stimulation (Fig. 2g;

**Fig. 1 | LPS increases sphingolipids via induction of Sptlc2 expression in M1-like macrophages. a** Experimental design in (**b**, **c**, **e**, **f**), and Figs. 2–4. **b** Heat map shows concentration $z$ scores of targeted metabolomics analysis (n = 6 biological replicates). More information in Supplementary Data 1. **c** Bar graphs showing RNA sequencing analysis of *Sptlc1-2*, *Asah1-2*, *Gba*, *Gba2*, *Smpd1-4*, *Sgpp1* and *Galc* in mouse BMDM after PBS or LPS treatment (n = 4 biological replicates). Original RNA sequencing dataset GSE140610 was previously published[9]. Cpm, counts per million. **d** Sphingolipid biosynthetic pathway. **e** Representative flow cytometry plots displaying the expression of CD38 and Egr2 in BMDM (live CD45 + CD11b + F4/80+ cells) under M0-like, M1-like or M2-like stimuli. Histograms and bar graphs show the Sptlc2 and ceramide mean fluorescence intensity (MFI) of M0-like, M1-like, and M2-like BMDM (n = 3 biological replicates). Gating in Supplementary Fig. 1a; independent experiment in Supplementary Fig. 1b. **f** Immunoblot analysis of Sptlc2 protein expression in BMDM from WT or *Lyz2*-cre mice after 20 h of treatment with PBS or LPS. Grp94 served as a loading control. Quantification was performed in FIJI.

Independent experiments in Supplementary Fig. 1e. **g** Histograms and bar graphs showing the expression of Sptlc2 in intraperitoneal and splenic macrophages (live CD45 + CD11b + CD3-B220-NK1.1-Ly6G-F4/80+ cells) from WT mice 3 h after PBS or LPS injection, and from WT or *Lyz2*-cre mice 3 h after LPS injection (n = 6 biological replicates). Data are presented as mean ± SD (**c**, **e**, **g**) and are pooled from two independent experiments (**b**, **g**). Statistical comparisons were performed with two-tailed unpaired Student's *t*-tests (**c**: all except Sptlc2 and Smpd1 and **g**: all except graph bottom left; data points were normally distributed), one-way analysis of variance (ANOVA) tests (**e**; for simultaneous comparisons of more than two groups), and two-tailed Mann-Whitney U tests (**c**: Sptlc2 and Smpd1 and **g**: graph bottom left; data points were not normally distributed). 9-week-old female and male (**b**, **e**–**g**) C57BL/6 mice were used. **a** Created with BioRender.com released under a Creative Commons Attribution-NonCommercial-NoDerivs 4.0 International license. Source data are provided as a Source Data file.

Supplementary Fig. 2h). Supplementation with sphinganine and with 3-KDS, which is rapidly metabolized to sphinganine[13,14], but not with sphingomyelin, cholesterol or sphingosine-1-phosphate, restored the percentages of FSC-A$^{high}$, SSC-A$^{high}$ cells among all *Lyz2*-cre BMDM to similar levels as in the WT group (Fig. 2g; Supplementary Fig. 2h). Together, these results suggested that *Sptlc2* deficiency decreases macrophage cell growth and metabolic fitness.

### Sptlc2 is required for LPS-induced recruitment of MyD88 to TLR4 in the cell membrane

To further examine how *Sptlc2* deficiency influenced LPS-induced signal transduction, we analyzed the expression of the exclusive M1-like macrophage marker CD38, which is known to be regulated by LPS via the transcription factor NF-κB, and the exclusive M2-like macrophage marker Egr2[15]. We detected significantly lower levels of CD38 after M1-like activation and higher levels of Egr2 after M2-like activation in *Sptlc2*-deficient macrophages than in *Sptlc2*-sufficient macrophages (Fig. 3a; Supplementary Figs. 1a; 3a), thus suggesting that *Sptlc2* deficiency partially reversed LPS-induced signal transduction and macrophage polarization. We further examined the protein levels of components in the LPS-TLR4-NF-κB signaling pathway. Levels of nuclear factor of kappa light polypeptide gene enhancer in B-cells inhibitor alpha (IκBα) decreased in *Sptlc2*-sufficient, but not -deficient BMDM early after LPS stimulation (Fig. 3b; Supplementary Fig. 3b, c). In line with these findings, phosphorylation of NF-κB p65 (Ser536), which has been associated with NF-κB signaling inhibition in macrophages[16–19], was markedly enhanced by *Sptlc2* deficiency (Fig. 3b; Supplementary Fig. 3b, c). In contrast to TIRAP, whose expression was not influenced by *Sptlc2* deficiency, MyD88 protein levels were lower in *Sptlc2*-deficient macrophages than in *Sptlc2*-sufficient macrophages after LPS stimulation (Fig. 3b; Supplementary Fig. 3b, c). Furthermore, flow cytometry confirmed that Sptlc2 was required for the complete LPS-induced upregulation of MyD88 protein levels (Fig. 3c; Supplementary Figs. 1a; 3d).

To test whether the reduced MyD88 expression levels in *Sptlc2*-deficient macrophages impaired the BMDM growth (Fig. 2g; Supplementary Fig. 2h), we overexpressed a WT and a L252P-mutated version of MyD88[20] in *Sptlc2*-sufficient and –deficient macrophages, but we did not observe a complete rescue of BMDM growth (Supplementary Fig. 3e, f). To examine whether Sptlc2 influenced MyD88 recruitment to TLR4 in the cell membrane, we performed confocal fluorescence microscopy. In *Sptlc2*-sufficient macrophages, LPS stimulation increased MyD88 protein levels, BMDM size, and enrichment in MyD88 at the cell membrane (Fig. 3d; Supplementary Fig. 3g). In contrast, LPS treatment scarcely altered MyD88 protein levels or recruitment to the cell membrane in the smaller *Sptlc2*-deficient macrophages (Fig. 3d; Supplementary Fig. 3g). Collectively, these results suggested that *Sptlc2* deficiency decreases the cell membrane recruitment of MyD88, thereby impairing the initiation of TLR4 signaling.

To evaluate whether the defect in the recruitment of MyD88 to the cell membrane in *Sptlc2*-deficient macrophages was responsible for the decrease in cell growth and M1-like macrophage marker expression, we retrovirally transduced macrophages with a vector for expression of MyD88 fused with an *N*-myristoylation (Myris) membrane-attachment signal sequence (Fig. 3e). *N*-myristoylation, a co-translational lipid modification specific to the alpha-amino group of an *N*-terminal glycine residue in proteins, enables membrane anchoring of these proteins[21,22]. Overexpression of MyrisMyD88 enhanced the membrane localization of MyD88 as well as the cell size of *Sptlc2*-deficient macrophages (Fig. 3f; Supplementary Fig. 3h). Furthermore, we performed flow cytometry to monitor the percentages of FSC-A$^{high}$, SSC-A$^{high}$ cells among the virally transduced macrophages, as demonstrated by the expression of a GFP reporter (Fig. 3g; Supplementary Fig. 3i). *Sptlc2*-deficient macrophages overexpressing the MigR1-GFP empty vector had lower percentages of FSC-A$^{high}$, SSC-A$^{high}$ large cells than *Sptlc2*-sufficient macrophages (Fig. 3h)—a phenotype like that observed in non-transduced macrophages (Fig. 2g, Supplementary Fig. 2h). Echoing the microscopy results (Fig. 3f), MyrisMyD88 overexpression increased cell size and CD38 expression of *Sptlc2*-deficient macrophages to a level comparable to that of *Sptlc2*-sufficient macrophages (Fig. 3h, i; Supplementary Fig. 3j, k), thus suggesting that Sptlc2 promotes the recruitment of MyD88 to the cell membrane and subsequent initiation of LPS-TLR4 signal transduction in macrophages.

### Sphinganine physically interacts with TLR4 signaling components

To better understand the mechanisms through which Sptlc2 promotes the recruitment of MyD88 to the cell membrane, we tested whether sphinganine, the metabolite restoring the cell size and morphology of *Sptlc2*-deficient macrophages, might enhance LPS-induced MyD88 translocation to the membrane. Because sphingolipids have been shown to directly interact with proteins in regulating signal transduction[23], we hypothesized that sphinganine might physically interact with proteins in the TLR4 signaling pathway. To test this hypothesis, we incubated macrophage lysate with biotinylated sphinganine, then performed precipitation with streptavidin-conjugated magnetic beads (Fig. 4a). We examined the region of the Coomassie-stained gel between 30 kDa and 40 kDa, where TLR4 adaptors would appear (Fig. 4b; Supplementary Fig. 4a). According to previous results (Fig. 3), we suspected TLR4 adaptors to be involved in sphingolipid-regulated TLR4 signaling. Subsequent mass spectrometry analysis revealed that the TLR4 adaptor protein MyD88 was pulled down together with biotinylated sphinganine (Fig. 4b; Supplementary Fig. 4a; Supplementary Data 2). We further confirmed, through western blotting, that sphinganine interacted with TIRAP and MyD88 (Fig. 4c; Supplementary Fig. 4b, c). The interactions between sphinganine and TIRAP and MyD88 were specific, as demonstrated by the lack of binding between sphinganine and several other proteins

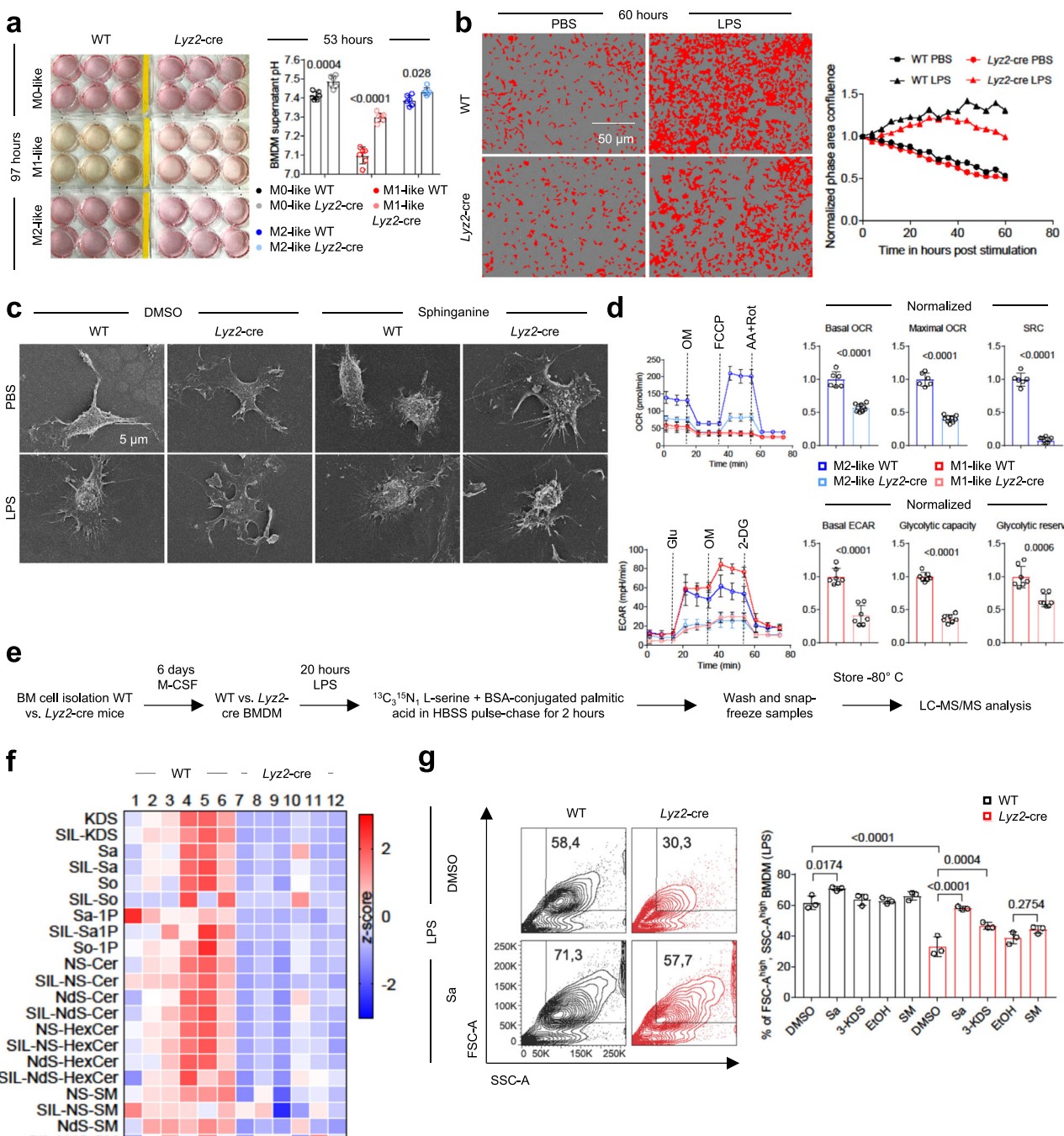

**Fig. 2 | *Sptlc2* deficiency decreases M1-like macrophage growth and this effect is reversed by sphinganine. a** Photographs of polarized WT or *Lyz2*-cre BMDM. Bar graphs show culture medium pH (n = 6 biological replicates). Independent experiment in Supplementary Fig. 2a. **b** Images and line graph illustrating confluency of PBS and LPS-treated WT or *Lyz2*-cre BMDM (n = 4 biological replicates; more information in Supplementary Fig. 2b, c). **c** Scanning electron microscopy images of WT or *Lyz2*-cre BMDM after DMSO or sphinganine and PBS or LPS treatment. Independent experiments in Supplementary Fig. 2d. **d** Line graphs showing changes in the oxygen consumption rate (OCR) (M1 WT: n = 10; M2 WT: n = 6; M1 *Lyz2*-cre: n = 10; M2 *Lyz2*-cre: n = 9 biological replicates), and the extracellular acidification rate (ECAR) of polarized WT or *Lyz2*-cre BMDM (M1 WT: n = 7; M2 WT: n = 8; M1 *Lyz2*-cre: n = 7; M2 *Lyz2*-cre: n = 11 biological replicates). Bar graphs show normalized basal OCR, maximal OCR, and spare respiratory capacity (SRC) (WT: n = 6; *Lyz2*-cre: n = 9 biological replicates) and basal ECAR, glycolytic capacity and glycolytic reserve (n = 7 biological replicates). Independent experiment in Supplementary Fig. 2e. OM oligomycin, FCCP carbonyl cyanide-p-trifluoromethoxyphenylhydrazone, AA antimycin A, Rot rotenone, 2-DG 2-deoxyglucose, Glu glucose. **e** Experimental design for metabolomics analysis in (**f**). **f** Heat map showing concentration of endogenous and stable isotope labeled (SIL) sphingolipids in LPS-polarized WT or *Lyz2*-cre BMDM (n = 6 biological replicates). **g** Dot plots and bar graph showing percentages of FSC-A^high, SSC-A^high WT or *Lyz2*-cre LPS-activated BMDM after sphingolipid supplementation (n = 3 biological replicates). Independent experiments in Supplementary Fig. 2h. Data are presented as mean ± SD (**a**, **d**, **g**) and pooled from 2 independent experiments (**f**). Statistical comparisons were performed with one-way ANOVA tests (**a**, **g**; for simultaneous comparisons of more than two groups), two-tailed unpaired Student's *t*-tests (**d**: all except glycolytic reserve; data points were normally distributed), and two-tailed Mann-Whitney U tests (**d**: glycolytic reserve; data points not normally distributed). 11-week-old female and male (**a–d**, **f**, **g**) C57BL/6 mice were used. Source data are provided as Source Data file. So sphingosine, Sa-1P sphinganine-1-phosphate, So-1P sphingosine-1-phosphate, Cer ceramide, HexCer hexosylceramide, NS non-hydroxy-fatty acid sphingosine, NdS non-hydroxy-fatty acid dihydro-sphingosine.

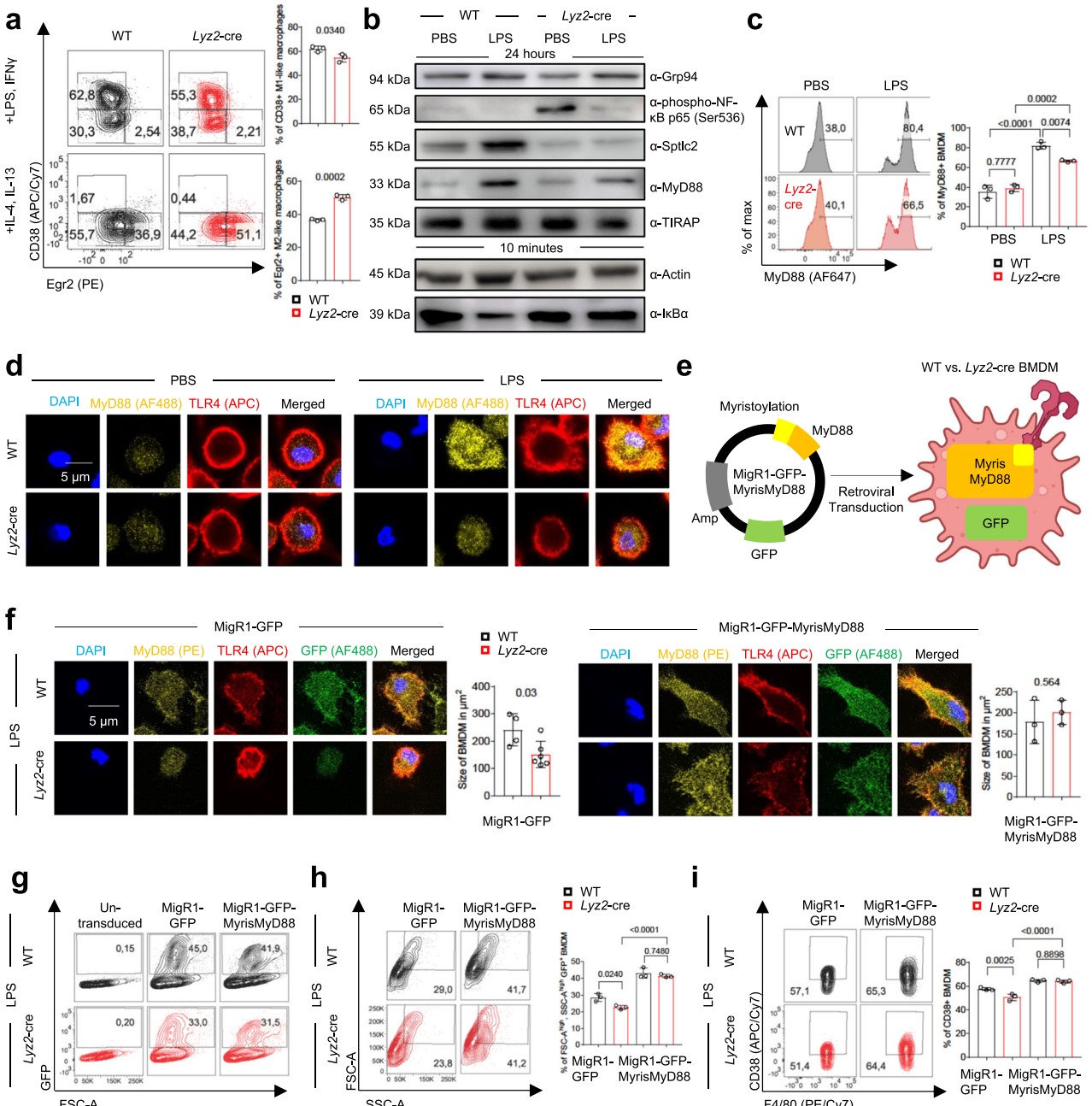

**Fig. 3 | Deficiency in *Sptlc2* mitigates the M1-like macrophage phenotype by preventing co-localization of TLR4 and MyD88 at the cell membrane and downstream NF-κB signaling. a** Histograms and bar graphs showing percentages of CD38+ and Egr2+ BMDM among all BMDM after M1- or M2-like activation, respectively (n = 3 biological replicates). Independent experiments in Supplementary Fig. 3a. **b** Immunoblot analysis of WT or *Lyz2*-cre BMDM after PBS or LPS stimulation. Independent experiments in Supplementary Fig. 3c. **c** Histograms and bar graph showing percentages of MyD88-expressing BMDM after PBS or LPS stimulation (n = 3 biological replicates). Results are pooled from 3 independent experiments with each 1 pair of mice; each point represents results from individual experiment. Independent experiment in Supplementary Fig. 3d. **d** Confocal fluorescence microscopy images, showing MyD88 and TLR4 in WT or *Lyz2*-cre BMDM after PBS or LPS stimulation. Independent experiments in Supplementary Fig. 3g. **e** Illustration of the experimental design in (**f**–**i**). Myris, myristoylation **f** Confocal fluorescence microcopy showing MyD88 and TLR4 in PBS-/LPS-treated WT or *Lyz2*-cre BMDM transduced with MigR1-GFP or MigR1-GFP-MyrisMyD88. Cell sizes were

visualized in bar graphs (WT MigR1-GFP: n = 4; *Lyz2*-cre MigR1-GFP: n = 6; MigR1-GFP-MyrisMyD88: n = 3 biological replicates). Bar graph data are pooled from 3 individual experiments (more images in Supplementary Fig. 3h). **g** Representative flow cytometry dot plots, showing transduction efficiency in LPS-activated WT or *Lyz2*-cre BMDM. More data in Supplementary Fig. 3i. **h** Flow cytometry dot plots and bar graphs, showing percentages of FSC-A^high, SSC-A^high GFP + LPS-stimulated BMDM (n = 3 biological replicates). Independent experiment in Supplementary Fig. 3j. **i** Dot plots and bar graphs showing percentages of GFP + CD38 + LPS-stimulated BMDM (n = 3 biological replicates). Independent experiment in Supplementary Fig. 3k. Data are presented as mean ± SD (**a**, **c**, **f**, **h**, **i**). Statistical comparisons were performed with two-tailed unpaired Student's *t*-tests (**a**, **f**; data points were normally distributed) and one-way ANOVA tests (**c**, **h**, **i**; for simultaneous comparisons of more than two groups). 9-week-old female and male (**a**–**d**, **f**–**i**) C57BL/6 mice were used. **e** Created with BioRender.com released under a Creative Commons Attribution-NonCommercial-NoDerivs 4.0 International license. Source data are provided as a Source Data file.

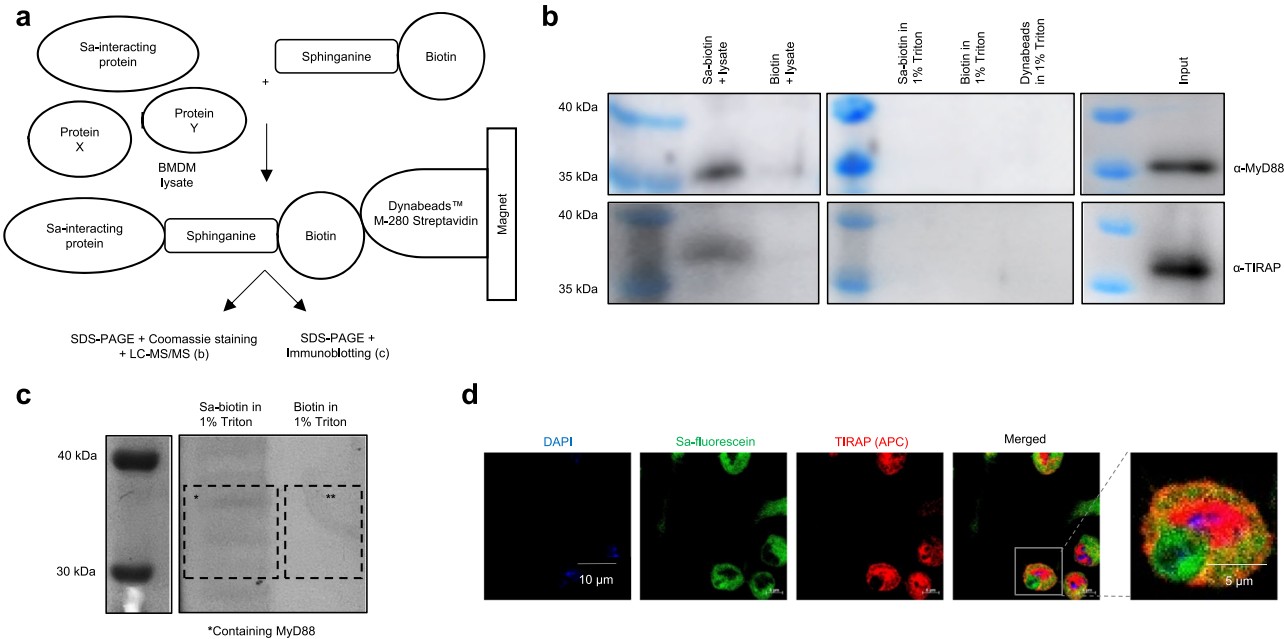

**Fig. 4 | Sphinganine physically interacts with the TLR4 adaptors TIRAP and MyD88. a** Experimental design in (**b**, **c**). Sphinganine (Sa)-interacting proteins were pulled down from WT BMDM lysate with sphinganine-biotin coupled to Streptavidin-Dynabeads with a magnet and identified through non-biased Coomassie staining (**b**) and an LC-MS/MS approach, and validated with a biased immunoblotting approach (**c**). **b** Proteins in the indicated Coomassie gel regions were identified by LC-MS/MS; MyD88 was only pulled down by sphinganine-biotin, but not control-biotin (see respective boxes; n = 1). Additional information is provided in Supplementary Data 2. Independent experiment in Supplementary Fig. 4a. **c** Immunoblot analysis of TIRAP and MyD88 after sphinganine-biotin and biotin

control pulldown. Sphinganine-biotin, biotin, and Dynabeads in lysis buffer (1% Triton X-100 in PBS) served as controls for the pulldown, and input samples validated the correct localization of the respective proteins on the membrane. Independent experiment in Supplementary Fig. 4c. **d** Confocal fluorescence microscopy images showing subcellular co-localization of supplemented sphinganine (Sa)-fluorescein and TIRAP in WT BMDM after PBS. Data are representative from four mice from three independent experiments (**d**); independent experiments in Supplementary Fig. 4e. 9-week-old female (**b**–**d**) C57BL/6 mice were used. Source data are provided as a Source Data file.

involved in signal transduction, such as mitogen-activated protein kinase kinase (MEK1/2), extracellular signal-regulated kinase (Erk1) and protein kinase B (Akt) (Supplementary Fig. 4d).

Furthermore, we cultured macrophages with fluorescently labeled sphinganine (sphinganine-fluorescein) and observed co-localization of TIRAP and sphinganine near the cell membrane (Fig. 4d; Supplementary Fig. 4e). Collectively, these findings suggested that sphinganine physically interacts with the TLR4 adaptor proteins TIRAP and MyD88.

### *Sptlc2* deficiency ameliorates LPS-induced sepsis symptoms

Excessive signaling and cytokine production through the LPS-TLR4-NF-κB axis in macrophages is considered a major driver of sepsis, which affects ~50 million people each year, among whom at least 11 million die[1,24]. Re-analysis of data from a previous study from Liepelt et al. indicated that patients with sepsis have higher expression levels of *Sptlc2* mRNA in CD14+ monocytes than healthy individuals[25] (Fig. 5a).

To test whether Sptlc2 might be required for LPS-induced inflammatory responses in vivo, we used an LPS-induced sepsis mouse model. We intraperitoneally injected LPS into mice with myeloid cell-specific *Sptlc2* deficiency or their wildtype littermates (Supplementary Fig. 5a). Half of the *Sptlc2*-sufficient mice showed a loss of movement and a hunched posture 6 h after LPS injection, whereas only ~20% of the *Sptlc2*-deficient mice showed these LPS-induced phenotypes (Fig. 5b, c; Supplementary Fig. 5b, c Supplementary Movies 1 and 2). Because plasma cytokine levels have been reported to peak within the first h after LPS exposure[26], we sacrificed mice 3–6 h after LPS challenge; by that time, no weight loss or differences in splenocyte numbers were observed in *Sptlc2*-sufficient or -deficient mice (Supplementary Fig. 5d, e). We performed flow cytometry to

analyze intraperitoneal macrophages and, other than after PBS injection, found fewer intraperitoneal macrophages in *Sptlc2*-deficient mice after LPS challenge (Fig. 5d, e; Supplementary Fig. 5f, g). Unlike after PBS injection, after LPS injection *Sptlc2*-deficient mice showed an increase in the percentages of intraperitoneal macrophages expressing the M2-like marker arginase-1 (Arg-1; Fig. 5f; Supplementary Fig. 5h, i). The plasma concentrations of the pro-inflammatory cytokines IL-12/IL-23 p40 and IL-12 p70 were higher in *Sptlc2*-sufficient than *Sptlc2*-deficient mice (Fig. 5g; Supplementary Fig. 5j). To gain a broader view of the influence of *Sptlc2* deficiency on LPS-induced inflammation in vivo, we used a cytokine array to monitor 40 cytokines, chemokines and acute-phase proteins in the plasma. *Sptlc2* deficiency caused a general decrease in inflammation-associated proteins (Fig. 5h; Supplementary Fig. 5k, l), including C5a, I-309 (CCL1), eotaxin (CCL11), IL-12 p70 and IL-23.

We explored the impact of *Sptlc2* deletion in a polymicrobial sepsis model using intraperitoneal cecal slurry injections[27,28] (Supplementary Fig. 6a). Similar as the LPS-induced sepsis model, cecal slurry injection increased Sptlc2 levels (Supplementary Fig. 6b) and *Lyz2*-cre mice exhibited an impaired M1-like and enhanced M2-like macrophage phenotype (Supplementary Fig. 6c). Plasma cytokine levels showed a declining trend in *Lyz2*-cre mice, even though not statistically significant (Supplementary Fig. 6d).

To test whether MyD88 recruitment to TLR4 might rescue inflammatory cytokine production in *Sptlc2*-deficient macrophages, we measured the levels of IL-12 p70 and IL-6 from the supernatants of in vitro generated LPS-/IFNγ-activated M1-like BMDM after over-expression of MyrisMyD88 (Fig. 5i; Supplementary Fig. 5m). Similarly to the in vivo cytokine array and ELISA results, *Sptlc2*-deficient macrophages overexpressing the MigR1-GFP empty vector had

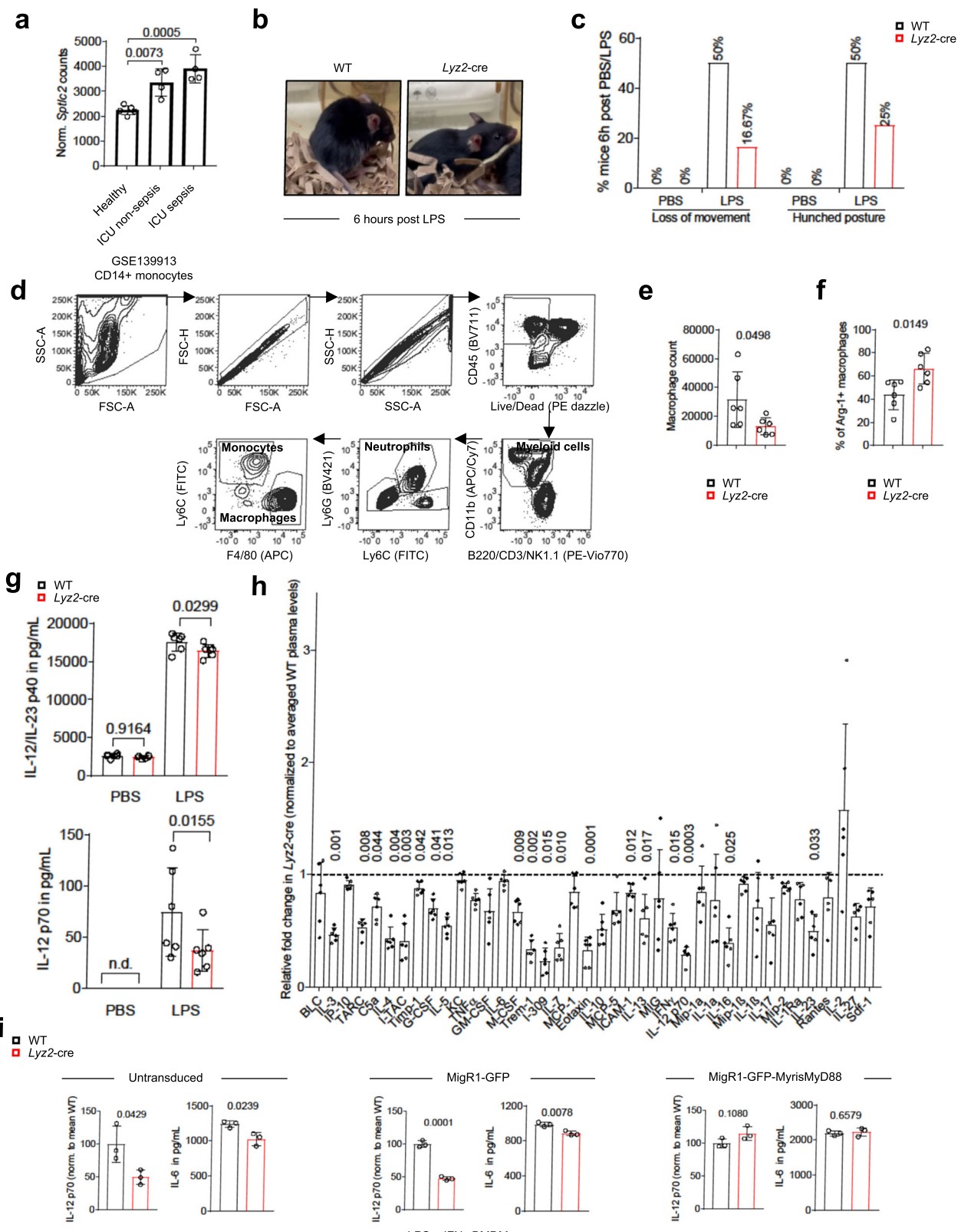

lower levels of IL-12 p70 and IL-6 than *Sptlc2*-sufficient macrophages (Fig. 5i; Supplementary Fig. 5m). In contrast, the concentrations of IL-12 p70 and IL-6 were comparable between *Sptlc2*-deficient macrophages overexpressing MyrisMyD88 and *Sptlc2*-sufficient macrophages overexpressing MyrisMyD88 (Fig. 5i; Supplementary Fig. 5m). Collectively, these results suggested that *Sptlc2* deficiency decreases LPS-induced inflammation in vivo and ameliorates sepsis

symptoms through attenuating MyD88-directed signal transduction (Fig. 6).

### *Sptlc2* deficiency decreases anti-tumor myeloid cell activity and increases tumor growth

Beyond the sepsis model, we used a B16 melanoma mouse model to test the role of Sptlc2 in regulating macrophage function in vivo

**Fig. 5 | *Sptlc2* deficiency mitigates LPS-induced sepsis symptoms and M1-like macrophage phenotype and cytokine production. a** Bar graph showing RNA sequencing analysis of *Sptlc2* in CD14+ monocytes from healthy volunteers without sepsis and intensive care unit (ICU) patients with sepsis (healthy: n = 5; ICU with or without sepsis: n = 4 biological replicates). Original RNA sequencing dataset GSE139913 was previously published[25]. **b, c** Representative images of WT or *Lyz2*-cre mice after LPS injection, and bar graphs illustrating the percentages of WT or *Lyz2*-cre mice with loss of movement and hunched posture (PBS: n = 3, LPS: n = 12 biological replicates). Independent experiment in Supplementary Fig. 5b, c. **d** Dot plots showing gating strategy used in (**e, f**). **e** Bar graph showing WT or *Lyz2*-cre macrophage counts after LPS injection (n = 6 biological replicates). Independent experiment in Supplementary Fig. 5g. **f** Bar graph showing percentages of Arg-1+ macrophages 6 h after LPS injection (n = 6 biological replicates). Independent experiment in Supplementary Fig. 5i. **g** Bar graphs, showing plasma IL-12/IL-23 p40 and IL-12 p70 levels 3 h after PBS or LPS injection (n = 6 biological replicates; PBS IL-

12 p70 below detection limit labeled n.d. not determined). Independent experiment in Supplementary Fig. 5j. **h** Bar graph showing pixel density of cytokine dots in *Lyz2*-cre plasma normalized to WT plasma. Black dashed line separates up- and down-regulated cytokines in *Lyz2*-cre mice; 3 h after LPS injection; (n = 6 biological replicates). Results are pooled from 2 independent experiments with each 3 pairs of mice. **i** Bar graph showing normalized IL-12 p70 and IL-6 levels in BMDM medium from 4 h LPS-/IFNγ-activated WT or *Lyz2*-cre BMDM, overexpressing MigR1-GFP or MigR1-GFP-MyrisMyD88 (n = 3). Independent experiment in Supplementary Fig. 5m. Data are presented as mean ± SD (**a, e–i**). Statistical comparisons were performed with one-way ANOVA tests (**a, g**; for simultaneous comparisons of more than two groups), two-tailed unpaired Student's *t*-tests (**e, f, h**: all except IL-4, G-CSF, Eotaxin and (**i**); data points were normally distributed) and Mann-Whitney U tests (**h**: IL-4, G-CSF, Eotaxin; data points were not normally distributed). 8-week-old female and male C57BL/6 mice were used (**b–i**). Source data are provided as a Source Data file.

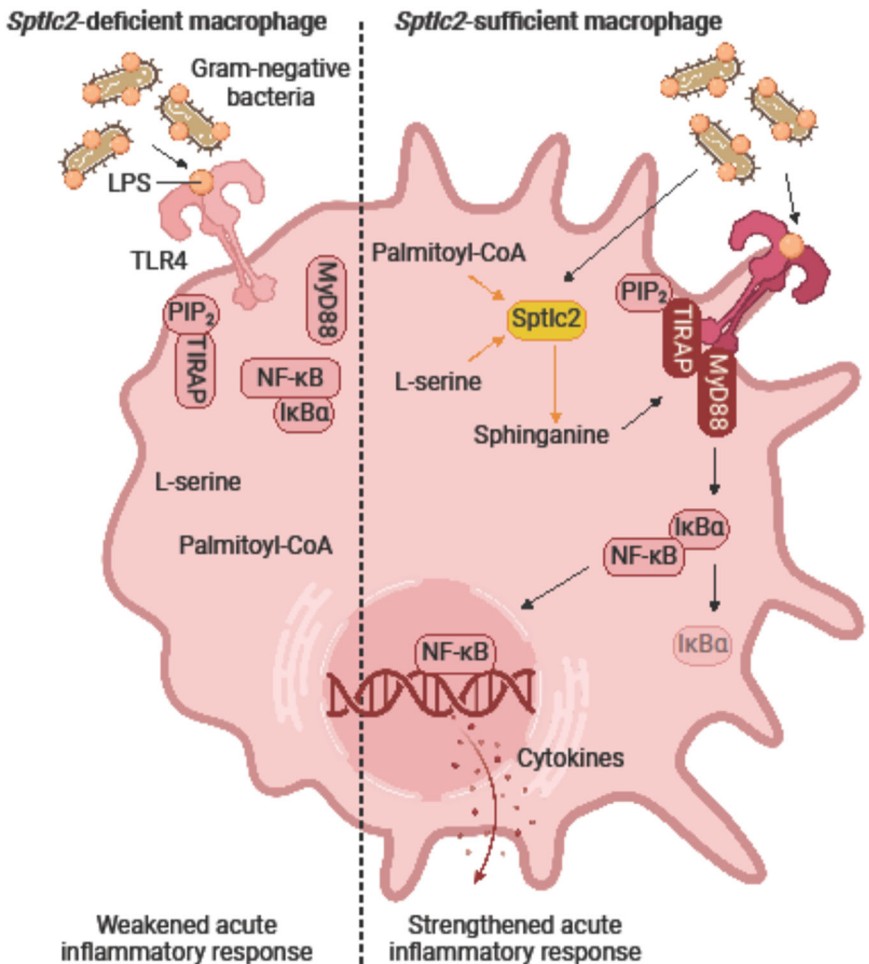

**Fig. 6 | Sptlc2-derived sphinganine is induced by LPS and recruits TLR4 adaptors in macrophages to promote inflammatory responses.** Illustration of the role of Sptlc2-derived sphinganine onto LPS-induced TLR4 signaling in macrophages. LPS induces Sptlc2-mediated sphinganine production and sphinganine interacts physically with TIRAP and MyD88 to recruit MyD88 to TLR4 in the macrophage membrane. This induces downstream signaling and results in M1-like activation-

associated morphologic changes and cytokine release, mediating a strengthened acute inflammatory response. In absence of Sptlc2, MyD88 is not recruited to the membrane, preventing activation of downstream signaling. Figure 6 created with BioRender.com released under a Creative Commons Attribution-NonCommercial-NoDerivs 4.0 International license.

(Supplementary Fig. 7a). In line with recently reported findings[29–35], LPS was readily detectable in tumors through immunoblotting and limulus amebocyte lysate assays (Fig. 7a; Supplementary Fig. 7b, c). Both the O55:B5 serotype and O111:B4 serotype of LPS were more enriched in B16 tumors than the spleen. The abundance of the O55:B5 serotype of LPS was comparable between tumor and skin tissue, and tumors had higher amounts of the O111:B4 serotype of LPS than the skin (Fig. 7a;

Supplementary Fig. 7b, c). Furthermore, tumor-associated macrophages (TAMs) expressed higher levels of Sptlc2 protein than splenic macrophages in a B16-F10 tumor model (Fig. 7b; Supplementary Fig. 7d, e). *Sptlc2* deficiency did not impact splenocyte or tumor-infiltrating leukocyte numbers (Supplementary Fig. 7f) but increased the tumor growth longitudinally and the tumor weight at the endpoint (Fig. 7c, d). A subset of dendritic cells also expressed lysozyme M[36].

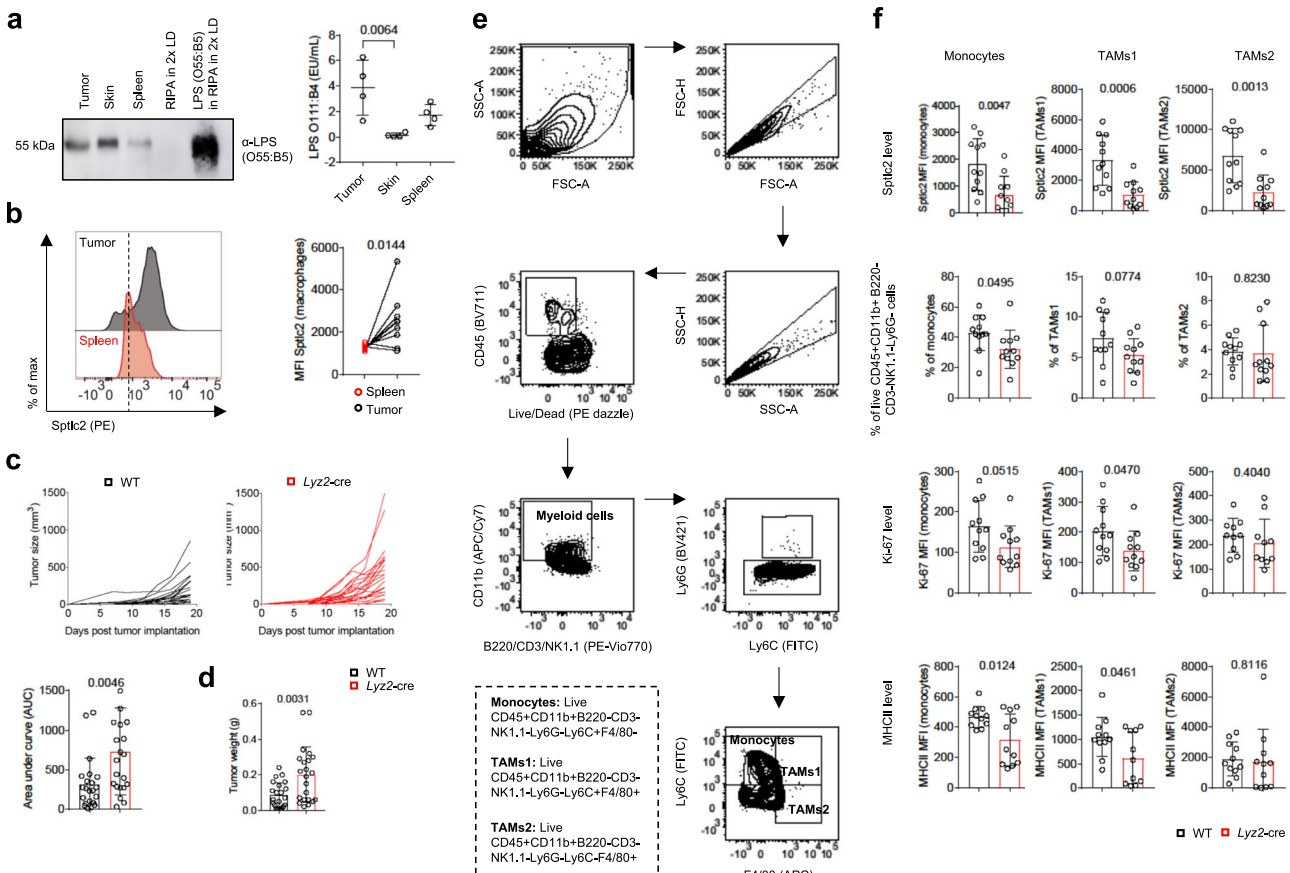

**Fig. 7 | *Sptlc2* deficiency increases B16-F10 tumor growth and weakens anti-tumor myeloid cell activity. a** Immunoblot analysis of LPS (O55:B5) in tumor, skin, and spleen lysates of B16-F10 tumor-bearing WT mice; LPS O55:B5 was used as a loading control (left). Scatter dot plot, showing LPS (O111:B4) concentrations in tumor, spleen, and skin tissues of B16-F10 tumor-bearing mice, measured with limulus amebocyte lysate assays (right) (n = 4 biological replicates). Independent experiments in Supplementary Fig. 7c. **b** Representative flow cytometry histograms and graph, showing normalized Sptlc2 MFI of macrophages from spleen or tumor tissues of the same mice (n = 10 biological replicates). Independent experiment in Supplementary Fig. 7e. **c** Line graphs, showing B16-F10 melanoma growth in WT or *Lyz2*-cre littermate mice over time after subcutaneous injection of 2 × 10⁵ B16-F10 melanoma cells (left). Bar graph, showing the area under the curve (AUC) of the tumor growth plots (n = 22 biological replicates). **d** Bar graph, showing the tumor weights of WT or *Lyz2*-cre mice at the endpoint on day 19 (n = 22). **e** Flow cytometry dot plots, showing the gating strategy used in (**f**) to identify monocytes (live

CD45 + CD11b + NK1.1-CD3-B220-Ly6G-Ly6C + F4/80- cells) and macrophages (TAMs1: live CD45 + CD11b + NK1.1-CD3-B220-Ly6G-Ly6C + F4/80+ cells; TAMs2: live CD45 + CD11b + NK1.1-CD3-B220-Ly6G-Ly6C-F4/80+ cells). **f** Bar graphs, showing Sptlc2, Ki-67 and MHCII expression levels in WT or *Lyz2*-cre monocytes, TAMs1 and TAMs2, and their percentages among of all live CD45 + B220-CD3-NK1.1-CD11b + Ly6G- cells from the tumor (n = 11 biological replicates). The data are pooled from two (**f**) or three (**c, d**) independent experiments, and are presented as mean ± SD (**a, c, d, f**); symbols with connecting lines highlight paired samples (**b**). Tumor growth curves were evaluated longitudinally, and the AUC was calculated for each tumor growth curve (**c**). Statistical comparisons were performed with one-way ANOVA tests (**a**; for simultaneous comparisons of more than two groups), two-tailed paired (**b**) or unpaired Student's *t*-tests (**d, f**; data points were normally distributed), and two-tailed Mann-Whitney U tests (**c**; data points were not normally distributed). 8-week-old female and male C57BL/6 mice were used (**a–f**). Source data are provided as a Source Data file.

*Sptlc2*<sup>Flox/Flox</sup> *CD11c*-cre mice, which had dendritic cell-specific *Sptlc2* deficiency, grew tumors of similar size, and the tumor weights did not significantly differ from those in wildtype littermates (Supplementary Fig. 7g), thus suggesting that the enhanced tumor growth observed in *Sptlc2*<sup>Flox/Flox</sup> *Lyz2*-cre mice was not due to nonspecific deletion of *Sptlc2* in dendritic cells.

To examine which myeloid cell types were affected by *Sptlc2* deficiency, we used flow cytometry to analyze myeloid cell subsets (Fig. 7e). The protein levels of Sptlc2 were relatively lower in *Lyz2*-cre monocytes, TAMs1, and TAMs2 (Fig. 7f). *Sptlc2* deficiency significantly decreased the percentages of monocytes but not TAMs2 (Fig. 7f). Moreover, we observed a trend toward relatively lower percentages of TAMs1 in *Sptlc2*-deficient mice, although this finding was not statistically significant (Fig. 7f). LPS stimulation induces macrophage proliferation and MHC expression via TLR4-induced NF-κB activity[37]. *Sptlc2* deficiency decreased macrophage proliferation and MHCII expression in monocytes and TAMs1 but not TAMs2 (Fig. 7f). In line with the decreased MHCII expression in monocytes and TAMs1 in vivo,

in the in vitro model, *Sptlc2*-deficient BMDM showed diminished MHCI<sup>high</sup>, CD86<sup>high</sup> expression and less potent induction of the expression of the T cell activation marker CD44 in the co-culture model of activated, LCMV gp33-41-pulsed BMDM and LCMV gp33-41-recognizing P14 CD8+ T cells (Supplementary Fig. 7h, i). These results were similar to the observation that *Sptlc2* deficiency decreased the percentages of tumor-infiltrating T cells among all live CD45+ cells and increased the percentages of T cells expressing the immune checkpoint molecules PD1 and TIGIT in the B16 melanoma model (Supplementary Fig. 7j, k). Collectively, these results suggested that Sptlc2 is required for monocyte and macrophage-mediated anti-tumor responses.

## Discussion

Our findings indicated that sphinganine physically interacts with TLR4 adaptor proteins; increases MyD88 recruitment to the cell membrane; and promotes LPS-induced cell growth, M1-like macrophage marker expression, and inflammatory cytokine production. Our work revealed

a role of sphingolipids in directly engaging in innate immune cell signal transduction. This study expands the traditional view of sphingolipids as building blocks that support biosynthesis and sheds new light on the multifaceted role of sphingolipids as signaling molecules in innate immunity.

Accumulating evidence has indicated the existence of a metabolic-inflammatory circuit, in which the bi-directional communication between metabolism and inflammation intricately links cellular metabolic pathways to inflammation-associated signal transduction. For example, inhibiting the biosynthesis of cholesterol sensitizes macrophages to di-cyclic nucleotides and enhances antiviral immunity[38]. Furthermore, inhibition of de novo fatty acid synthesis attenuates IL-17-producing helper T (Th17) cell differentiation and limits Th17 cell-mediated inflammation[39]. Our study indicated that LPS-induced de novo biosynthesis of sphingolipids. Conversely, inhibition of de novo synthesis of sphingolipids antagonized TLR4 adaptor protein recruitment to the cell membrane and impaired downstream signal transduction, thereby providing an example of pattern recognition coupling lipid metabolism and inflammation and inducing a self-enhancing metabolic-inflammatory circuit in macrophages.

One open question is which chemical groups in sphinganine are responsible for the interaction with TLR4 adaptor proteins. In this study, we used biotinylated sphinganine and found that sphinganine physically interacted with MyD88 and TIRAP. The biotin was conjugated to the end of the N-acyl chain of sphinganine. Therefore, the N-acyl chain of sphinganine was occupied and was unlikely to be available for interacting with proteins. This hypothesis was supported by the absence of sphinganine-interacting proteins pulled down with agarose beads coupled to the head group of sphinganine via Schiff base formation. Further biochemical analyses are required to more comprehensively test the possibility that the polar head group, rather than of the N-acyl chain of sphinganine, mediates the interaction with adaptor proteins.

TLR4 signaling is a promising target for immunotherapies. Intra-tumor LPS injections activate anti-tumor immune cells and decrease tumor growth[40], but the strong pro-inflammatory properties of LPS result in severe adverse effects[41]. The need for clinically applicable TLR4 signaling modulators remains unmet. The current study revealed a Sptlc2-sphinganine pathway, which may serve as a new target for fine-tuning TLR4 signaling in diverse disease settings. For example, inhibition of Sptlc2-dependent sphinganine production would decrease sepsis-associated inflammation. This strategy echoes that in a previous study showing that modulating the metabolism of gangliosides, a group of sphingolipids, decreases LPS-induced intestinal inflammation[42]. While previous studies indicated that gangliosides function as inhibitors of TLR4 signaling[5,43], our research reveals a pivotal role for sphinganine in facilitating complete TLR4 activation. Parallel with the established sphingolipid-rheostat model of specific ceramides triggering apoptosis[44] and sphingosine-1-phosphate promoting cell survival[45], considering the role of gangliosides in TLR4 signaling[5,43], our findings complement to another example in which interconversion of bioactive sphingolipids determines cell fate. Those findings together with our results suggest that sphingolipid-based approaches may create new therapeutic opportunities for pharmacological modulation of sphingolipid metabolism and amelioration of LPS-induced inflammation.

Lipid rafts play an important role in TLR4 complex assembly and internalization[46–49]. Sphingolipids are a component of lipid rafts. Sptlc2-mediated sphingolipid synthesis promotes TLR4 signal transduction by enhancing the recruitment of the TLR4 adaptor protein MyD88 to the cell membrane. We do not exclude the possibility that Sptlc2 might strengthen TLR4 signaling through additional mechanisms, such as enlarging the pool of sphingolipids and enhancing the formation of lipid rafts.

TLR4 signaling is activated by additional ligands beyond LPS, such as palmitic acid[50]. Our previous study has suggested that palmitic acid is enriched in the tumor microenvironment[51]. We detected two serotypes of LPS in tumors, but we cannot exclude the possibility that palmitic acid and additional ligands of TLR4 might require Sptlc2-mediated sphingolipid synthesis to activate the signal transduction downstream of TLR4. Furthermore, TLR4-independent pathways, which are induced or inhibited through Sptlc2 deficiency, might potentially have contributed to the phenotype of the tumor model. We are aware of two limitations of this study. First, although we showed that Sptlc2 deficiency markedly decreased overall sphingolipid levels, whether Sptlc2 deficiency might induce compensatory mechanisms (such as increasing transport of extracellular sphingolipids) is unknown. Second, we detected two serotypes of LPS in tumors. Whether other serotypes of LPS might also exist in the tumor model used in the current study is also unknown.

Overall, our findings revealed that LPS induces the expression of Sptlc2 and promotes de novo sphingolipid synthesis in macrophages. Sphinganine, in turn, bolsters TLR4 signal transduction by physically interacting with TLR4 adaptor proteins and recruiting MyD88 to the cell membrane. This study suggests that macrophages rewire sphingolipid metabolism in response to LPS stimulation and uncovers a molecular link between lipid metabolism and pattern recognition in both physiological and pathological settings.

## Methods

### Mice
This research complies with all relevant ethical regulations and all studies were performed in accordance with DKFZ regulations after approval by the German regional council at the Regierungspräsidium Karlsruhe (G266-19, G255-16, G236-21). Mice were maintained in the German cancer research center (DKFZ) specific-pathogen-free facility. The mice were housed in a 12 h on (7 am–7 pm), 12 h off light cycle with room temperature of 20–24 °C and humidity of 45–65%. The $Sptlc2^{Flox/Flox}$ mice were kindly provided by Professor Xian-cheng Jiang (SUNY Downstate Medical Center, New York, USA) via Professor Vishwa Dixit (Yale University, New Haven, USA) and Professor Susan Kaech (Salk Institute for Biological Studies, La Jolla, USA)[52]. Exon 1 of Sptlc2 is flanked by two LoxP sites and is excised after crossing with a Cre-expression mouse strain. $Lyz2$-cre mice[53] and $CD11c$-cre mice[54] were from the Jackson Laboratory. For generation of myeloid cell- or dendritic cell-specific Sptlc2 knockout mice, $Sptlc2^{Flox/Flox}$ mice were bred with $Lyz2$-cre or $CD11c$-cre mice, respectively. Offspring from this breeding, which had one allele of Sptlc2 floxed, and which had a $Lyz2$- or $CD11c$-cre, were again bred with $Sptlc2^{Flox/Flox}$ mice to generate $Sptlc2^{Flox/Flox}$ $Lyz2$-cre or $CD11c$-cre mice. Wildtype littermate mice were used as controls. We used sex-matched and age-matched (7–17 week-old) mice for each individual experiment. In rare cases, mice with fighting wounds were excluded from the experimental analysis. The sample collection and processing were not performed in a blinded manner.

### BMDM generation
With sterile scissors and tweezers, the femur and tibia were isolated from mouse legs and cut open. Bone marrow was rinsed from the bones with IMDM (Gibco, #21980-032), containing 10% FBS (Sigma-Aldrich, #F7524) and 1% Penicillin-Streptomycin (Pen-Strep; Gibco, #15140122) with a 27 G needle and pressed through cell strainers (mesh size 70 μm, Greiner). Red blood cells were lysed with ACK lysis buffer (LIFE Technologies, #A1049201) for 1 min and the reaction was stopped by the addition of four volumes of IMDM, containing 10% FBS and 1% Pen-Strep. After centrifugation ($805 \times g$, 2 min), remaining cells were resuspended in IMDM, containing 10% FBS and 1% Pen-Strep, and passed through a nylon mesh (Sefar AG) to obtain a single cell suspension. Cells were counted with Trypan Blue (Sigma-Aldrich, #T8154) and a Neubauer counting chamber (Thermo Fisher Scientific). If not otherwise indicated, $3 \times 10^6$ BM cells in 2 mL IMDM, containing 10% FBS

and 1% Pen-Strep supplemented with 10 ng/mL M-CSF (Miltenyi-Biotech, #130-101-700)/well of a six-well plate (TPP) were seeded and cultured for 5–6 days in an incubator (Heraeus) at 5% $CO_2$, 37 °C. On day 3, the medium was changed and, if indicated, sphingolipids were supplemented in the medium (3-KDS (d18:0), Cayman Chemical, #Cay24380; ceramide (C16), Avanti Polar Lipids, #860516P; Cholesterol, Sigma-Aldrich, #C8667; Sphinganine (d18:0), Cayman Chemical, #Cay10007945; Sphingomyelin, Enzo Life Sciences, #BML-SL135; Sphingosine-1-phosphate, Avanti Polar Lipids, #860492P). On day 6 or 7, differentiated BMDM were polarized with PBS (Gibco, #14040-091) toward M0-like, with 100 ng/mL LPS (Sigma-Aldrich, #LPS25) alone (indicated as +LPS) or in combination with 50 ng/mL IFNγ (PeproTech, #315-05) (indicated as M1-like) toward M1-like or with 10 ng/mL IL-4 (BioLegend, #574304) and 10 ng/mL IL-13 (BioLegend, #575904) toward M2-like macrophage phenotypes. Cell scrappers (Costar®) were used to remove BMDM from culture dishes for subsequent analyses.

## Targeted metabolomics analysis

BMDM were polarized to M0-like, M1-like or M2-like macrophages for 24 h. The polarization medium of BMDM was collected, centrifuged ($805 \times g$, 2 min) to remove floating cells and stored at −80 °C. BMDM were washed in 2 mL ice-cold 0.9% NaCl solution and, after removal of the washing solution, 0.7 mL 0.9% NaCl solution was added in each well. BMDM from two wells were gently scraped and transferred into pre-cooled 1.5 mL microcentrifuge tubes. BMDM were counted for later normalization. After centrifugation ($805 \times g$, 2 min), the supernatant was completely removed, and pellets were snap-frozen and stored at −80 °C. Cell pellets and 100 µL supernatant were submitted to Metabolomics Core Technology Platform of the University of Heidelberg, Heidelberg, Germany. Cell extracts for downstream metabolomics analyses were prepared with the 75% EtOH/MTBE protocol[55]. In total, 1019 metabolites, including 14 small molecules and 25 different lipid classes, were analyzed with the MxP Quant 500 XL kit (Biocrates life sciences ag, #21469.2) according to the manufacturer's protocol. In brief, 10 µl of cell culture supernatants or cell extracts was pipetted into a 96 well-plate containing internal standards and dried under a nitrogen stream with a positive pressure manifold (Waters). Subsequently, 50 µl of a 5% phenyl isothiocyanate solution was added to each well to derivatize amino acids and biogenic amines. After 1 h incubation at room temperature, the plate was dried again. For extraction of the metabolites, 300 µl 5 mM ammonium acetate in methanol was pipetted onto each filter and incubated for 30 min. The extract was eluted into a new 96-well plate with positive pressure. For further LC-MS/MS analyses, 150 µl of the extract was diluted with an equal volume of water. For FIA-MS/MS analyses, 10 µl extract was diluted with 490 µl of FIA solvent (provided by Biocrates). For preparation of the FIA XL plate, 10 µl of cell culture supernatants or cell extracts was pipetted into a second 96 well-plate containing internal standards and dried under a nitrogen stream. Subsequently, 300 µl extraction solvent (19 mg ammonium acetate in 50 mL methanol) was added to each well, incubated 30 min, and eluted into a new 96-well plate with positive pressure. For FIA XL-MS/MS analyses, 50 µl extract was diluted with 450 µl of FIA solvent. After dilution, LC-MS/MS and FIA (XL)-MS/MS measurements were performed. Chromatographic separation was performed with a UPLC I-class PLUS (Waters) system coupled to a SCIEX QTRAP 6500+ mass spectrometry system in electrospray ionization (ESI) mode. Data were generated with the Analyst (Sciex) software suite and transferred to the WebIDQ software (Biocrates) which was used for further data processing and analysis. All metabolites were identified with isotopically labeled internal standards and multiple reaction monitoring (MRM) with optimized MS conditions, as provided by Biocrates. For quantification, either a seven-point calibration curve or a one-point calibration was used, depending on the metabolite class.

## Cell lysis and immunoblotting

BMDM were lysed with the RIPA lysis buffer system (Santa Cruz Biotechnology, #sc-24948) or 1% Triton X-100 in PBS (Bio-Rad Laboratories, #161-0407; supplemented with 200 mM PMSF, protease inhibitor in DMSO (Sigma-Aldrich, #D8418) and 100 mM sodium orthovanadate in water from the RIPA lysis buffer system) when sphinganine-biotin pulldown was performed before immunoblotting. For detection of LPS by immunoblotting, we performed normalization by loading the same mass of tissue (in mg) per lane (3 µL of the RIPA lysis buffer system per 1 mg spleen, tumor, or skin tissue). LPS from *E. coli* O55:B5 (Sigma-Aldrich, #L2637,) was used as a positive control. For the other immunoblots, 2 µl lysate was used to measure the protein concentration with a Pierce™ BCA Protein Assay kit (Thermo Fisher Scientific, #23227). Total protein was heated at 95 °C in 2× or 4× Laemmli sample buffer (Bio-Rad Laboratories, #1610737, #161747 and #J61337) supplemented with 2-mercaptoethanol (Gibco, #31350010) for 10 min. Then proteins were equally loaded and resolved with 12% SDS-PAGE (75 V for 15 min and then 60–90 min at 120 V, until the blue indicator ran to the edge of the gel; PowerPac™ Basic Power Supply, Bio-Rad). Proteins were subsequently transferred onto PVDF membranes (Merck Millipore, #ISEQ00010; 100 V, 2 h). The membranes were blocked with 5% BSA (Sigma-Aldrich, #A9647) in PBS supplemented with 0.1% Tween® 20 (Promega, #H5152; PBST) for 2 h at room temperature, then incubated overnight at 4 °C with anti-Sptlc2 (Origene, #TA319780), anti-Grp94 (Cell Signaling Technology, #20292, D6X2Q), anti-Bcl-2 (Cell Signaling Technology, #3498, D17C4), anti-phospho-NF-κB p65(Ser536; Cell Signaling Technology, #3033, 93H1), anti-MyD88 (Cell Signaling Technology, #4283, D80F5), anti-TIRAP (Cell Signaling Technology, #13077, D6M9Z), anti-IκBα (Cell Signaling Technology, #4812, 44D4), anti-LPS (Thermo Fisher Scientific, #MA5-41631, C6), anti-Erk1 (Cell Signaling Technology, #4372), anti-Akt (Cell Signaling Technology, #9272) or anti-MEK1/2 (Cell Signaling Technology, #9122) on a shaker. The PVDF membrane was washed three times (10 min each) with PBST, then incubated with HRP-conjugated anti-mouse (BioLegend, #405306, Poly4053) or anti-rabbit (BioLegend, #406401, Poly4064) secondary antibodies on a shaker at room temperature for 2 h. The membrane was developed with the ECL method (Cytiva, #RPN2235), and the data were collected with a Fusion system (FX6 Edge, Vilber Lourmat). We quantified the band intensities in the ImageJ program (NIH). For re-staining proteins on the same membrane, we stripped the membranes by washing in 0.2 M NaOH (Carl Roth, #T135.1) for 20 min, then washed them three times (10 min each) with PBST. Subsequently, membranes were again blocked and stained.

## Flow cytometry

Fc receptors were blocked for at least 15 min on ice with Ultra-LEAF™ Purified anti-mouse CD16/32 antibody (BioLegend, #101330) in PBS supplemented with 1% FBS (FACS buffer) to prevent nonspecific antibody binding. For surface antigen staining, cells were incubated in FACS buffer with fluorophore-conjugated antibodies for at least 30 min on ice. Live/DEAD Fixable Dead Cell Stain kits (Thermo Fisher Scientific, #L34972, #L34963) were used to exclude the dead cells. When only surface antigens were stained, fixation buffer containing 4% paraformaldehyde (BioLegend, #420801) was used. For staining of intracellular antigens, cells were fixed and permeabilized on ice (Thermo Fisher Scientific, #00-5523-00) for at least 30 min, and intracellular antigens were stained directly or via primary and secondary antibodies. Samples were washed and run on an LSR II or Canto flow cytometer and analyzed in FlowJo software (v10, FlowJo).

## Flow cytometry antibodies

The antibodies utilized in this study were recommended by the manufacturer for use with the mouse species. Their specificity and accuracy were validated in each experiment by comparing the observed molecular weight against a protein ladder. Antibodies used for

detection of the following antigens: CD11b (BioLegend, #101212 and #101226, M1/70; 1:400), F4/80 (Miltenyi Biotech, #130-116-547, #130-118-327 and BioLegend, #123114, BM8 and REA126; 1:400), CD38 (Bio-Legend, #102728 and Miltenyi Biotech, #130-128-224, REA616 and 90; 1:400), Egr2 (Miltenyi Biotech, #130-114-256, REA869; 1:100), CD45 (BioLegend, #103147, 30-F11; 1:400), B220 (Miltenyi Biotech, #130-110-711, REA755; 1:400), CD3 (Miltenyi Biotech, #130-116-530, REA641; 1:400), CD4 (BioLegend, #100406; 1:400), NK1.1 (Miltenyi Biotech, #130-116-533, PK136; 1:400), Ly6G (Miltenyi Biotech, #130-119-902, REA526), Ly6C (BioLegend, #128006, HK1.4; 1:200), IL-12 (Miltenyi Biotech, #130-102-163, REA136; 1:200), Ki-67 (BioLegend, #652424, 16A8; 1:400), MHCI (BioLegend, #111518, KH95; 1:400), MHCII (BioLegend, #107626 and #107608, M5/114.15.2; 1:400), CD45 (BioLegend, #103149, 30-F11; 1:400), CD44 (BioLegend, #103057, IM7; 1:200), TIGIT (BioLegend, #142104, 1G9; 1:100), CD8a (BioLegend, #100804, 5H10-1; 1:400), CD86 (BioLegend, #105008, GL-1; 1:400), Granzyme B (BioLegend, #372210, QA16A02; 1:400), IgG (BioLegend, #406421; 1:400), TNFα (BioLegend, #506306; 1:200), PD-1 (BioLegend, #135216; 1:400) and CD16/32 (BioLegend, #101330, 93; 1:500). Sptlc2 (rb; polyclonal; 1:400), ceramide (Enzo Life sciences, #ALX-804-196-T050, ms; MID 15B4; 1:100), MyD88 (rb; #D80F5; 1:100) and Arg-1 (Cell Signaling Technology, #93668, rb; D4E3M; 1:100) were detected via secondary anti-rb-AF647 (BioLegend, #406414; 1:200), -AF488 (BioLegend, #406416; 1:200) or -PE (BioLegend, #406421, all poly4064; 1:200), or anti-mouse-BV421™ (#406517, RMM-1; 1:200) staining.

## pH measurement
pH values were determined with a pH meter (pH 50 Benchmeter, VioLab). Between measurements, the electrode was carefully rinsed with ddH$_2$0 and, to avoid temperature-induced pH changes, the pH was determined immediately after plates were removed from the incubator.

## Confluency analysis
In a 12-well plate (TPP), $2.5 \times 10^5$ BM cells per well were differentiated to BMDM. On day 3 of BMDM cultivation, cells were placed in an Incucyte S3 (Sartorius) instrument, and confluency images were recorded every 4 h for a total of 6 days. On day 6 of BMDM cultivation (3 days after confluency measurement was started), BMDM were PBS-/LPS-stimulated. Confluency analysis was performed in Incucyte2028B software.

## Scanning electron microscopy
In each well of a 12-well plate (TPP), $1 \times 10^5$ or $2.5 \times 10^5$ BMDM were grown on glass coverslips; on day 3, the medium was supplemented with DMSO, or sphinganine was added. On day 6, BMDM were PBS- or LPS-polarized and submitted to the Electron Microscopy Group (W230), German Cancer Research Center (DKFZ), Heidelberg, Germany. Cells grown on glass coverslips were fixed with 2% buffered glutaraldehyde (100 mM cacodylate at pH 7.2, supplemented with 1 mM MgCl$_2$ and 1 mM CaCl$_2$), post-fixed with 1% OsO$_4$ and dehydrated in a graded ethanol series (Sigma-Aldrich, #32205). After critical point drying (Balzers 030), samples were sputter coated with Au/Pd 80:20 (Balzers 050) for immediate SEM observation. Scanning electron micrographs were taken with a Zeiss Auriga SEM (Carl Zeiss) at 5 kV acceleration voltage and a ~2 mm work-distance with an in-lens detector for secondary electrons.

## Seahorse extracellular flux analysis (Seahorse analysis)
BMDM were grown and detached with 0.25% trypsin-EDTA (Gibco, #25200) 1 day before the assay and counted, and $7 \times 10^4$ or $1 \times 10^5$ BMDM were seeded into culture plates (Agilent) for Mito or Glyco stress tests, respectively, and polarized. The Seahorse sensor cartridges from the Seahorse XFe96 FluxPak (Agilent Technologies, #102416-100) were hydrated overnight at 37 °C, according to the manufacturer's instructions. We seeded BMDM in seahorse base

medium supplemented with either 2 mM glutamine (Sigma-Aldrich, #G7513), 10 mM D-(+)-glucose (Sigma-Aldrich, #G8644) and 1 mM sodium pyruvate (Sigma-Aldrich, #P5280) to perform Mito stress testing or with 2 mM glutamine to perform Glyco stress testing. During Mito stress testing, the following compounds were injected into the culture plate sequentially: oligomycin (Sigma-Aldrich, #495455; 20 μM), FCCP (Cayman Chemical, #15218-50; 1 μM), and a mixture of antimycin A (Sigma-Aldrich, #A8674; 2 μM) and rotenone (Cayman Chemical, #13995-5; 2 μM). During Glyco stress testing glucose (20 mM), oligomycin (10 μM), and 2-deoxy-D-glucose (Cayman Chemical, #14325; 50 mM) were injected into the culture plate sequentially. The oxygen consumption and extracellular acidification rates were recorded automatically with a Seahorse XFe96 flux analyzer (Agilent).

## Metabolic tracer labeling, lipid extraction, and analysis by LC-MS/MS
LPS-polarized (Sigma-Aldrich, #LPS25) BMDM were washed in cold PBS, scraped (Costar®) from the culture dishes (TPP), re-suspended in HBSS (Gibco, #14175; 1% FBS) and counted. The same cell numbers were pulse-chased with tracer-labeled 500 μM $^{13}C_3^{15}N_1$ L-serine (Sigma-Aldrich, #608130) and 10 μM BSA-conjugated palmitic acid (Sigma-Aldrich, #P0500, in NaOH) for 2 h at 37 °C, 5% CO$_2$. Every 30 min, the cells were gently mixed. The cells were then washed twice in cold PBS, snap-frozen on dry ice, and stored at −80 °C. The cell pellets were submitted to the Lipid Pathobiochemistry Group (A411), German Cancer Research Center (DKFZ), Heidelberg, Germany, and subjected to lipid extraction and analysis by LC-MS/MS. According to ref. [56] 100 μL of a methanolic solution containing a mix of internal lipid standards [C12-3-ketosphinganine, C20-sphinganine, C20-sphingosine, D7-sphingosine (d18:1), D7-sphinganine (d18:0), sphingosine (d20:1) 1-phosphate, Cer(d18:1/14:0), Cer(d18:1/19:0), Cer(d18:1/25:0), Cer(d18:1/ 31:0), 1-desoxy-Cer(d18:0/12:0), GlcCer(d18:1/14:0), GlcCer(d18:1/19:0), GlcCer(d18:1/25:0), GalCer(d18:1/31:0), SM(d18:1/12:0), SM(d18:1/17:0), SM(d18:1/py), and SM(d18:1/31:0)] was added and further suspended in 400 μL chloroform:methanol (1:1).

The suspension was incubated in an ultrasound bath for 15 min at 37 °C and treated meanwhile 3 times for 3 min with ultrasound with 2 min breaks inbetween. The suspension was centrifuged at room temperature at $13000 \times g$ for 3 min and 450 μL of the clear supernatant was collected with a Hamilton syringe into a clean glass vial. The residual pellet was mixed with additional 450 μL of chloroform:methanol:water (10:10:1) and 10 μL of acetic acid, treated like before and the second supernatant was added to the first. The pooled extract was dried with a gentle nitrogen gas stream at 37 °C and finally taken up in 200 μL methanol:water (95:5) and subjected to UPLC-ESI-MS2 analysis. To detect sphingomyelins in the absence of phosphatidylcholines an aliquot of the samples was dried and treated with 25% aqueous ammonia:methanol (400 μL: 400 μL) at 80 °C for 5 h. After drying samples with a gentle nitrogen stream at 40 °C, samples were taken up in 200 μl 95% methanol for LC-MS2 analysis. Sample aliquots of 10 μL were injected onto Aqcuity I class UPLC (Waters) equipped with a CSH C18 reversed phase column (100 × 2.1 mm, 1.7 μm, Waters), which was heated to 60 °C. Non-acylated sphingoid bases were separated with gradient starting with 65% solvent A (50% Methanol and 50% water) and 35% solvent B (99% isopropanol and 1% methanol) both supplemented with 10 mM ammonium formate and 0.1% formic acid. After 0.2 min solvent B was increased to 42% at 0.4 min, to 70% at 4 min, and to 95% at 10 min. It was kept at 95% until 11 min and then further increased to 99% at 11.5 min. Thereafter solvent B was decreased back to 35% at 12 min and kept for another 2 min at 35% before the next sample was injected. Ceramides, Hexosylceramides and sphingomyelins were separated with a gradient starting with 57% solvent A and 43% solvent B. After 0.2 min solvent B was increased to 50% at 0.4 min. Thereafter this gradient was identical to the above

gradient except for going back to 43% B at the end for 2 min before the next sample was injected. Lipids were detected with a tandem mass spectrometer (Xevo TQ-S from Waters) in multi-reaction monitoring (MRM) mode by collision induced dissociation after ionization with electrospray. KDS was monitored with the transition of the protonated molecular ion $[M+H]^+$ to the fragment $[M+H-CH_2O]^+$ (at 14 eV), sphinganine and sphingosine with the transitions $[M+H]^+ \rightarrow [M+H-(CH_2O+H_2O)]^+$ (at 12 and 8 eV), sphingosine-1 phosphate with the transition $[M+H]^+ \rightarrow [M+H-(H_3PO_4+H_2O)]^+$ (at 8 eV), sphinganine-1 phosphate with the transition $[M+H]^+ \rightarrow [M+H-(H_3PO_4)]^+$ (at 12 eV), dihydroceramides with the transitions $[M+H]^+ \rightarrow [M+H-(FA)]^+$ (at 25-29 eV), ceramides with the transitions $[M+H]^+ \rightarrow [M+H-(FA+H_2O)]^+$ and $[M-OH]^+ \rightarrow [M+H-(FA+H_2O)]^+$ (at 22-24 eV), hexosyldihydroceramides with the transitions $[M+H]^+ \rightarrow [M+H-(FA+C_6H_{10}O_5)]^+$ (at 46-48 eV), hexosylceramides with the transitions $[M+H]^+ \rightarrow [M+H-(FA+Hexose)]^+$ and $[M-OH]^+ \rightarrow [M+H-(FA+Hexose)]^+$ (at 43-47 eV), and sphingomyelins and dihydrosphingomyelins with the transitions $[M+H]^+ \rightarrow [C_5H_{15}NO_4P]^+$ (35 eV). De novo synthesized SLs were discriminated from steady-state SLs by incorporation of $^{13}C_3,^{15}N_1$-stable isotope labeled L-serine leading to an n+3 mass shift of the corresponding molecular ions and a corresponding n+2/n+3 mass shift of the product ions in MRM mode. Data were processed using MassLynx software, whereas mass spectrometric peaks were quantified according to their peak area ratio with respect to the internal standard using TargetLynx software (both v 4.1 SCN 843) both from Waters Corporation (Manchester, UK). Subsequently, quantification of ceramides, hexosylceramides, and sphingomyelins was adjusted to the length of the acyl chain, and dihydro(hexosyl)ceramide quantification was further adjusted by a factor calculated between the intensities of external ceramides and dihydroceramid standards of the same concentration.

### Fluorescence staining of BMDM and confocal microscopy
PBS-washed 6-day-old BMDM were incubated for at least 15 min in PBS supplemented with 5% FBS and Ultra-LEAF™ purified anti-mouse CD16/32 antibody to prevent nonspecific antibody binding. For surface antigen staining, TLR4-APC (Thermo Fisher Scientific, #MTS510) was added for 1–2 h on ice on a shaker, and cells were fixed with fixation buffer containing 4% paraformaldehyde and permeabilized with eBioscience permeabilization buffer. Intracellular antigens, TIRAP (Cell Signaling Technology; rb; #D6M9Z) and MyD88 (Cell Signaling Technology; rb; #D80F5) were stained overnight at 4 °C and, as indicated, secondary antibody staining (Cell Signaling Technology; anti-rb #7074 or anti-mouse #7076) of intracellular antigens was performed for 1–2 h on ice on a shaker. Nuclei were stained for 5 min with DAPI (Sigma-Aldrich, #268298) in eBioscience permeabilization buffer, and samples were mounted on glass slides. Images were recorded under a confocal microscope (LSM710, Zeiss, ZEN Black Software) and analyzed with ZEN Blue Software and the NIH ImageJ program.

### LPS-induced sepsis model
LPS (150 µg in 150 µL PBS) from *E. coli* O111:B4 (Sigma-Aldrich, #L2630) or control PBS was intraperitoneally injected into mice. Sepsis symptoms were monitored. At the indicated times, mice were sacrificed, and ear biopsies were collected for genotyping, blood was collected for plasma isolation, and spleens and intraperitoneal flush samples were isolated. For intraperitoneal flushing, PBS supplemented with 2% FBS was injected into the intraperitoneum with a 27G needle and syringe, the mouse abdomen was gently massaged, and intraperitoneal flush samples were collected. The intraperitoneal cells were washed and processed in downstream applications.

### Cecal slurry-induced sepsis model
The cecal slurry was kindly provided from Prof. Dr. Hans-Reimer Rodewald, DKFZ, Heidelberg[27,28]. Cecal slurry (5 µL/g body weight) in PBS or control PBS was intraperitoneally injected into WT and *Lyz2*-cre

mice. Mice were sacrificed 3 h after injection and samples were isolated as described in the LPS-induced sepsis model.

### Cytokine array
Each membrane was probed with plasma from a different PBS- or LPS-injected (Sigma-Aldrich, #L2630) mouse, and the cytokine array (R&D systems, #ARY006) was performed according to the manufacturer's instructions. Images of the membranes were recorded with the ECL method in the Fusion system, and dot pixel densities were quantified in the NIH ImageJ program.

### ELISA
We used plasma of PBS- or LPS-injected (Sigma-Aldrich, #L2630) mice or supernatant of PBS- or LPS-stimulated (Sigma-Aldrich, #LPS25) BMDM and performed ELISA according to the manufacturer's instructions (R&D systems, #DY419, #DY2398, #DY406). Optical densities at 450 nm were determined in a microplate reader (Infinite M1000 Pro, TECAN). For the wavelength correction, readings at 540 nm were subtracted from readings at 450 nm.

### Sphinganine-biotin pulldown assays
Approximately 1 mg total BMDM protein lysate in 400–700 µL 1% Triton X-100 in PBS per reaction was incubated with 10 µg sphinganine-biotin (Echelon Biosciences, #S-110B) or control biotin (Sigma-Aldrich, #B4501) in EtOH for 14 h on a rotator at 4 °C. Subsequently, Dynabeads™ M-280 Streptavidin (Thermo Fisher Scientific, #11206) were added for 9 h, then pulled down with a magnet. Immobilized proteins were washed three times in PBST and eluted from the magnetic beads by boiling in 6× SDS sample buffer for 10 min at 95 °C. Proteins were run on 12% SDS gels and immunoblotted. In parallel, gels were washed three times in ddH2O (5 min each), stained with Coomassie Brilliant Blue R-250 solution (G-Biosciences, #786-498) for 1 h, or until all bands of the un-stained protein ladder became visible, and de-stained in Coomassie Brilliant Blue De-Staining solution (G-Biosciences, #786-499). Visible gel bands were cut from the gel with a scalpel (area from 30–40 kDa) and sent in 0.2% NaN3 in ddH2O to the proteomics core facility of the European Molecular Biology Laboratory, Heidelberg, Germany. Samples were subjected to in-gel digestion with trypsin[57]. Peptides were extracted from the gel by sonication for 15 min, and this was followed by centrifugation and supernatant collection. A solution of 50:50 water-acetonitrile, 1% formic acid was added for a second extraction, the samples were again sonicated for 15 min and centrifuged, and the supernatant was pooled with the first extract. The supernatants were dried down and reconstituted in 10 µL 4% acetonitrile and 1% formic acid in water, and analyzed with LC-MS/MS. An UltiMate 3000 RSLC nano LC system (Dionex) equipped with a trapping cartridge (µ-Precolumn C18 PepMap 100, 5 µm, 300 µm i.d. × 5 mm, 100 Å) and an analytical column (nanoEase™ M/Z HSS T3 column 75 µm × 250 mm C18, 1.8 µm, 100 Å, Waters) was coupled directly to an Orbitrap Fusion™ Lumos™ Tribrid™ Mass Spectrometer (Thermo Fisher Scientific) with the Nanospray Flex™ ion source in positive ion mode. Trapping was performed with a constant flow at 30 µL/min of 0.05% trifluoroacetic acid in water for 4 min. Subsequently, peptides were eluted via the analytical column at a constant flow rate of 0.3 µL/min with running solvent A (0.1% formic acid in water, 3% DMSO) with an increasing percentage of solvent B (0.1% formic acid in acetonitrile, 3% DMSO) from 2% to 8% in 1 min; 8% to 23% for 39 min; 23% to 38% in another 5 min; an increase in B to 85% in 4 min; and re-equilibration back to 2% B for 6 min. The peptides were introduced into the Orbitrap Fusion Lumos via a Pico-Tip Emitter 360 µm OD × 20 µm ID; 10 µm tip (CoAnn technologies) and an applied spray voltage of 2.4 kV. The capillary temperature was set at 275 °C. Full mass scans were acquired from 350 to 1500 m/z in profile mode in the Orbitrap with a resolution of 120000. The maximum fill time was 250 ms. The instrument was operated in data-dependent acquisition mode, and MSMS scans were

acquired in the ion trap in rapid mode, with a full time as high as 35 ms and AGC target set to standard. A normalized collision energy of 30 was applied, and the activation type was HCD. MS$^2$ data were acquired in centroid mode. The raw data were processed with MaxQuant (v1.6.3.4)[58] and searched against the UniProt *Mus musculus* database (*mus musculus*, 17,114 entries, July 2022). Carbamidomethyl (C) was set as a fixed modification, and acetyl (*N*-terminal) and oxidation (M) were set as variable modifications. The mass error tolerance for the full scan MS spectra was set to 20 ppm for MS/MS spectra to 0.5 Da. No more than two missed cleavages were allowed. For protein identification, a minimum of one unique peptide with a peptide length of at least 7 amino acids and a false discovery rate below 0.01 were required at the peptide and protein levels. Calculation of iBAQ values was enabled[59]. The raw output files of MaxQuant (proteinGroups.txt–file) were processed with the R programming language (ISBN 3-900051-07-0). As a quality filter, only proteins quantified with at least two unique peptides were allowed. A total of 1136 proteins were identified for sphinganine-biotin, and 565 proteins were identified for biotin samples. Raw iBAQ values were used in the analysis (columns starting with"iBAQ"). Missing values were imputed with the known method with the Msnbase package[60]. Differential abundance was evaluated by computation of the respective ratio of iBAQ signals.

## Cloning and overexpression

The restriction enzymes SgfI (New England BioLabs, #R0630) and MluI (New England BioLabs, #R3198) in CutSmart buffer (New England BioLabs, #B7204) were used for 4 h at 37 °C to excise *MyD88* from a commercially obtained *MyD88*-containing plasmid (2 μg; OriGene, #MR227105) and to linearize the plasmid MigR1-GFP (2 μg; modified to include BglII, SgfI, AscI, RsrII, MluI and XhoI restriction sites). After the addition of 6× TriTrack DNA loading dye (Thermo Fisher Scientific, #R1161), the digestion products were run on a 1% agarose gel, from which both MigR1-GFP and MyD88 were gel extracted according to the manufacturer's instructions (Qiagen, #28706). The DNA concentrations were determined with a NanoDrop spectrophotometer (Nano-Drop™ One, Thermo Fisher Scientific). With the T4-ligation system, the *MyD88* sequence (8 ng, 24 ng, and 40 ng) was shuttled overnight at 4 °C into the MigR1-GFP backbone (60 ng) to generate MigR1-GFP-MyD88. To generate a constitutively active variant of MyD88, a point mutation was introduced into the TIR domain of MyD88[20]. By replacing the thymidine nucleotide at position 755 with a cytosine nucleotide, the codon CTG (encoding leucine, L) was changed to CCG (encoding proline, P). By doing so, the amino acid leucine at position 252 was substituted with proline, resulting in an L252P mutant, referred to as MyD88$^{L252P}$. The MigR1-GFP-MyD88$^{L252P}$ plasmid was derived from the MigR1-GFP plasmid through a PCR using the customized MyD88$^{L252P}$ forward primer (5′ GGTGTCCAACAGAAGCGACCGATTCCTATTAAATA CAAG 3′) and the MyD88$^{L252P}$ reverse primer (5′ CTTGTATTTAATAG GAATCGGTCGCTTCTGTTGGACACC 3′) from Eurofins (Luxembourg, Luxembourg). The PCR product was agarose gel purified, and for removal of parental DNA, an overnight DpnI (New England BioLabs, #R0176S) digest at 37 °C was set up. The product of this reaction was amplified through the transformation of DH5α competent cells (Invitrogen, #18265017) and DNA was purified from individual colonies. A control digest of 2 μL purified DNA using SgfI and MluI restriction enzymes was performed, followed by agarose gel electrophoresis. Positive clones showing a band at ~900 bp representing the MyD88$^{L252P}$ and 7 kbp representing the backbone were sent for Sanger sequencing to Eurofins (Luxembourg, Luxembourg). Sequencing results were run in BLAST to ensure that MyD88 was L252P-mutated. To generate plasmids for the MyrisMyD88 overexpression, we designed a gene-specific primer pair for MyD88 (forward primer: 5′ GAAGAT CTATGGGCAGCAGCAAGAGCAAGCCCAAGGACCCCAGCCAGAGGAT GTCTGCGGGAGACCC 3′ and reverse primer: 5′ CCGCTCGAGT

CAGGGCAGGGACAAAGCCTTGG 3′), which contained protective bases (5′ GA 3′ in the forward primer and 5′ CCG 3′ in the reverse primer), restriction sites for BglII (5′ AGATCT 3′ in the forward primer) and XhoI (5′ CTCGAG 3′ in the reverse primer), the myristoylation signal sequence (underlined) and a *MyD88* sequence (5′ ATGTCTGCGGGA GACCC 3′ in the forward primer and TCAGGGCAGGGACAA AGCCTTGG 3′ in the reverse primer). We used these primers together with 10 ng MigR1-GFP-MyD88 plasmid in a PCR reaction with the Phusion® High-fidelity DNA polymerase (New England BioLabs, #M0530, #M0531) system (98 °C 30 s; 25× (98 °C, 10 s; 55–70 °C gradient, 45 s; 72 °C, 2 min); 72 °C, 10 min; and 4 °C hold; Bio-Rad PCR machine). Subsequently, 6× TriTrack DNA loading dye was added, and the PCR products were run on a 1% agarose gel, gel extracted as described above and digested with BglII (TaKaRa, #1021 A) and XhoI (TaKaRa, #1094 A) in 10× H buffer (TaKaRa, #SD0007) overnight at 37 °C. Empty MigR1-GFP plasmid was simultaneously digested and, after addition of 6× TriTrack DNA loading dye, run again on a 1% agarose gel, from which both the PCR product and plasmid were gel extracted. DNA concentrations were determined, and the PCR product was ligated into the linearized plasmid with the T4 ligase system (New England BioLabs, #M0201, #B0202) overnight at 4 °C. Subsequently, 5 μL of the ligation product MigR1-GFP-MyrisMyD88 was added to 50 μL DH5α cells for 30 min on ice, then introduced into bacterial cells via heatshock transformation (42 °C, 60–90 s). Bacteria were kept on ice for 1 min and then added to 250 μL plain LB medium and incubated for 1 h at 37 °C with shaking (10 × *g*). Subsequently, 200 μL was plated onto ampicillin agar plates (Sigma-Aldrich, #A5354) and cultivated overnight at 37 °C. On the next day, clones were picked and used to inoculate LB medium supplemented with ampicillin, and the cultures were grown overnight. Subsequently, Miniprep (Qiagen, #27106) was performed according to the manufacturer's instructions, and the DNA concentration was determined. Control digests with XhoI and BglII in 10× H buffer and subsequent visualization of the cut DNA fragments in a 1% agarose gel and Sanger sequencing (Eurofins) confirmed that the MyrisMyD88 sequence was correctly inserted into MigR1-GFP. MigR1-GFP-MyrisMyD88 or control plasmids (MigR1-GFP-MyD88, MigR1-GFP) and eco-helper plasmid were transfected into HEK293T cells (provided from Prof. Dr. Susan Kaech, Salk Institute for Biological Sciences, La Jolla, USA) via X-tremeGENE HP DNA transfection according to the manufacturer's instructions (Roche, #6366244001). On the next day, viral particles from the supernatant were used for retroviral transduction of 3-day-old BMDM by spin infection (1 h, 32 °C, 1500 *g*). Transduction efficiency was monitored according to GFP expression under a microscope (Zeiss) and in flow cytometry. Transduced BMDM were polarized on the next day and used for flow cytometry and confocal microscopy analyses.

## B16-F10 tumor model

B16-F10 melanoma cells (provided from Prof. Dr. Hanspeter Pircher, Max Planck Institut für Immunbiologie und Epigenetik, Freiburg, Germany) were cultured in cell culture flasks (TPP) in DMEM (Gibco, #41965) supplemented with 10% FBS, 1% Pen-Strep and 200 μg/mL G418 (Carl Roth, #0239.3). With a 27 G needle and syringe, $2 \times 10^5$ B16-F10 cells in 100 μL RPMI (Gibco, #21875) were subcutaneously injected into the flanks of mice, and tumor growth was measured every 3 days with a caliper (Twin-Cal, Tesa Technology). At day 19 or earlier, when the human endpoint criteria were reached, the mice were sacrificed, and tumor, spleen, skin, and ear samples were collected. Tumors were weighed (ADB, Kern). Dounce homogenizers were used for cell isolation from skin pieces for LPS immunoblotting and limulus amebocyte lysate assays. Tumor samples were pressed through cell strainers (Greiner), red blood cells were lysed with ACK lysis buffer, and single-cell suspensions of tumor samples were passed through nylon mesh (Sefar) and used in subsequent analyses.

## Limulus amebocyte lysate assays

We used tumor, spleen, and skin lysates from B16-F10 tumor-bearing mice, prepared as described in the cell lysis and immunoblotting section, using $3\,\mu L$ of the RIPA lysis buffer system per 1 mg spleen, tumor, or skin tissue. Limulus amebocyte lysate assays (Thermo Fisher Scientific, #A39552) were performed according to the manufacturer's instructions. Optical densities at 405 nm were determined with a microplate reader (Infinite M1000 Pro, TECAN).

## BMDM: CD8+ T cell co-cultivation assays

BMDM were grown, detached with cell scrapers (Costar®) on day 6, washed in cold PBS, and collected in 50 mL tubes (TPP). The cells were spun down ($805 \times g$, 2 min) and counted, and $1 \times 10^5$ cells were seeded with or without LCMV gp33-41 (MedChemExpress, #HY-P1569; 10 ng/μL) and 100 ng/mL LPS (Sigma-Aldrich, #LPS25) into 96-well flat-bottom plates (TPP) in 200 μL IMDM (10% FBS, 1% Pen-Strep) per well, and cultured overnight at 37 °C and 5% $CO_2$. Furthermore, on day 6, 96-well U-bottom plates (TPP) were coated with 0.5 μg/μL anti-CD3 (BioLegend, #100253) in PBS for at least 1 h at 37 °C under 5% $CO_2$. A spleen from a P14 mouse (provided from Prof. Dr. Hanspeter Pircher, Max Planck Institut für Immunbiologie und Epigenetik, Freiburg, Germany) was isolated and passed through a cell strainer (Greiner), red blood cells were lysed with ACK lysis buffer, and single-cell suspensions were generated with nylon mesh (Sefar). CD8a+ T cells were isolated by negative selection with a CD8a+ T cell isolation kit (Miltenyi, #130-104-075) according to the manufacturer's instructions. CD8+ T cells were seeded to 96-well U-bottom plates (TPP) in RPMI supplemented with 10% FBS, 1% Pen-Strep, 1% MEM NEAA (Gibco, #11140-035), 0.1% 2-mercaptoethanol and 0.2 μg/μL anti-CD28 antibody (BioLegend, #102116) in PBS, and cultured overnight at 37 °C under 5% $CO_2$. On the next day, CD8a+ T cells were washed in PBS and added to the PBS-washed BMDM in a 1:5 ratio of BMDM to T cells for co-cultivation for 3 days at 37 °C under 5% $CO_2$. Flow cytometry staining and analysis were performed as described above.

## Quantification and data analysis

We used GraphPad Prism (v7.05) to perform statistical analysis. When comparing two groups, we first determined whether the data points were normally distributed (D'Agostino & Pearson normality test, Shapiro-Wilk normality test or KS normality test). Statistical analysis of normally distributed data was performed with the two-tailed Student's $t$ test. Statistical analysis of data points that were not normally distributed was performed with the two-tailed Mann-Whitney U test (also known as the Wilcoxon rank sum test). Simultaneous comparisons of more than two groups were performed with ANOVA, as indicated in the figure legends. In all cases, $P < 0.05$ was considered statistically significant. Sample sizes are indicated in the figure legends. Data are presented as mean ± SD, or as specified in the figure legends. Unless indicated differently, measurements were taken from distinct samples.

## Reporting summary

Further information on research design is available in the Nature Portfolio Reporting Summary linked to this article.

# Data availability

The previously published sequencing data are available online at Gene Expression Omnibus (GSE139913: https://www.ncbi.nlm.nih.gov/geo/query/acc.cgi?acc=GSE139913 and GSE140610: https://www.ncbi.nlm.nih.gov/geo/query/acc.cgi?acc=GSE140610). The metabolomics and mass spectrometry data generated in this study are provided in the supplementary and source data. Source data are provided with this paper.

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

## Acknowledgements

We thank J. Schwarz, M. Rettl, and F. Stein at the Proteomics Core Facility (PCF) at European Molecular Biology Laboratory, Heidelberg, Germany for proteomic analysis; D. Krunic and M. Brom at the DKFZ Light Microscopy Core Facility for microscopy support; M. Volz at the Lipid Patho-biochemistry Group (DKFZ) for help in LC-MS/MS sample preparation. We thank the Metabolomics Core Technology Platform of the University of Heidelberg for support with metabolite quantification. We thank our colleagues at the DKFZ flow cytometry and animal housing facility for their assistance with our work. Some illustrations were created in BioRender.com (full license). Guoliang Cui is supported by a CRI Lloyd J. Old STAR Award (#3914; G.C.), a Helmholtz Young Investigator Award (#VH-NG-1113; G.C.), an EMBO Young Investigator Award, an Exploration grant of the Boehringer Ingelheim Foundation (BIS), the German Research Foundation (DFG, #CU375/5-1, #CU375/5-2, #CU375/7-1, #CU375/9-1; all to GC and #259332240/RTG2099 to G.C., V.U., and J.U.), the German Cancer Aid Foundation (DKH, #70113343 and #70114224; all G.C.), the Helmholtz Zukunftsthema Ageing and Metabolic Programming (AMPro, #ZT0026; G.C.), HI-TRON Kick-Start Seed Funding (#HITR-2021-08; G.C.), the Hector Foundation (#M20102; G.C.) and an ERC Consolidator Award (#101045416; G.C.).

## Author contributions

M.H. and G.C. designed the experiments. M.H. performed most of the biological experiments. M.H. prepared samples and R.S. performed lipid mass spectrometry. K.R. and M.H. performed scanning electron microscopy. M.H. prepared samples and G.P. performed targeted metabolomics. A. Madi assisted M.H. in re-analysis of previously published RNA sequencing data and in vivo mouse models. A. Madi, S.M., J.W., A. Mieg, K.M., N.K., D.S., F.B., and L.F. assisted with some experiments. M.H. and G.C. analyzed the biological data. M.H. and G.C. wrote the manuscript

with input from the other authors. V.U. and G.C. funded the work. J.U., V.U., and G.C. supervised the work.

## Funding

## Competing interests
G.C. receives research funding from Bayer AG and Boehringer Ingelheim, which has no direct relevance to the findings presented in this study. The remaining authors declare no competing interests.
