## [Peer Review File · Nature Communications]

Sphinganine Recruits TLR4 Adaptors in Macrophages and Promotes Inflammation in murine models of sepsis and melanomaREVIEWER COMMENTS

Reviewer #1 (Remarks to the Author):

Reviewer comments to the MS entitled 'Sphinganine membrane-anchors TLR4 adaptors in macrophages to promote inflammation'

Submitted to Nat. Comm. NCOMMS-23-28052-T

by Hering M, Madi A, Sandhoff R, Ma S, Wu J, Mief A, Richter K, Mohr K, ten Bosch N, Stichling D, Poschet G, Umansky V and Cui G

The MS is presenting impressive results and well written, covering biological phenomena up to elaborating molecular mechanisms to unravel a novel role of the sphingolipid sphinganine, which was previously held to function as a building block for other sphingolipids, especially sphingolipid-1-phosphate and ceramides, at the interfaces between pattern recognition as a function of the innate immune system and subsequent host response on the level of gene expression and release of proinflammatory cytokines. Experiments were well performed and described, statistical were adequately performed. Despite the impressive and interesting findings, there are some major and minor points of criticism prior to debate the acceptance of the MS to improve the quality of that.

Major points:

- Administration of LPS is not a sepsis model (Buras JA et al 2005). To improve the quality of the MS and to expand the outreach of your impressive results for clinical research, you have to perform a comparison of *sptlc2*-deficient mice with wt in an established model of polymicrobial sepsis.
- the role of cholesterol and ceramides in your experimental setting is unclear. In both conditions, infection and (sterile) inflammation, these two compounds are critically involved in assembly of membrane embedded proteins/receptors, relevant for PAMP-triggered signal transduction.
- sphingosine-1-phosphate is the most important product of metabolism of sphingosine and essential for a broad range of immune response mechanisms (membrane stabilization etc. etc.) without providing data on S1P concentrations. There is a lack of convincing your arguments.

Minor points:

- Nomenclature of endotoxin subtypes throughout the MS with an alphabetical character 'O' and not a digit 'zero'. <>
- in graphical abstract the term 'sepsis symptoms' is misleading, 'immune response' or anything else is more appropriate. <>
- the alphabetical characters N and C determining the terminus of a protein must be given in italic typeface <>

Reviewer #2 (Remarks to the Author):

This is a fairly complex paper about the role of sphingolipids in TLR4 mediated signaling. In particular, the authors found that LPS induces the production of various sphingolipids, particularly

sphinganine. which helps recruit MyD88 to the cell membrane to form a complex with TIRAP and the cytoplasmic portion of TLR4. The generated a conditional KO of Sptlc2, a key component of the sphingolipid biosynthetic pathway. Mutant cells deficient in Sptlc2 had a much diminished inflammatory response to LPS compared with Sptlc2 expressing cells, and the authors present evidence that this may be due to decreased MyD88 recruitment. Overall, Sptlc2 deficiency decreased inflammatory responses to injected LPS. Finally, Sptlc2 deficiency reduced the activity of macrophages against a melanoma tumor line in vivo, which they posited might be due to decreased responses to LPS within the tumor.

While I learned a lot reading this paper and feel it makes a nice contribution to our knowledge about sphingolipid biosynthesis in LPS responses, I was struck by a few things. First, the paper is difficult to read. Many of the sentences are very long and complex, when simpler language would suffice. This may be due to the fact that the authors are not native English speakers, and is not meant to be a criticism. However, the paper would be improved with editing. Second, I think the authors sometimes overreach in drawing conclusions. An example of this is the experiment where they express MyD88 in mutant myeloid cells. MyD88 did, in fact, alter the phenotype of Sptlc2-deficient macrophages towards that of wild type macrophages, but this could be explained in several ways and they chose only one explanation for the data. I was particularly surprised by their LPS "sepsis" model: rather than perform the studies as is usually done (ie, to inject increasing doses of LPS to induce partial lethality), they measure hunching and loss of movement. These are difficult criteria to assess. I assume that the reason they did not examine mortality was related to an ethical concern, but the authors would benefit by stating this if it is true. Measuring cytokine and other inflammatory markers is certainly an acceptable alternative to lethal LPS experiments, but in this regard, the differences between cytokines in the KO vs wild-type macs was not impressive, even if significant statistically. I suggest that these experiments would benefit from a time course and/or a dose response experiment. The authors should recognize that 6 hours may be rather late to be measuring cytokines. For example, TNF peaks about 1.5 hours after LPS challenge and would certainly have been worth measuring. Finally, LPS injection does not equal "sepsis" (this view of LPS challenge was discarded many years ago), and should not be referred to as such.

While I find the anti tumor experiments to be interesting, I think the authors might consider removing them from this manuscript and focusing on LPS signaling. The experiments on the anti tumor effect should be enlarged upon, and could easily be part of a followup manuscript.

Reviewer #3 (Remarks to the Author):

Hering et al. sought to investigate the role of bioactive lipid sphinganine in mediating toll-like receptor activation in macrophages. The authors employed a variety of approaches, along with M1 and M2 macrophage differentiation, lipidomic analysis, and different pulldown experiments to demonstrate that sphinganine is essential for the expression of the TIR adaptor MyD88 and its

partner TIRAP, as well as their recruitment to the cell membrane and subsequent activation of the NF- κ B pathway upon LPS activation. Additionally, they observed that LPS stimulation induces the expression of sphinganine, promoting cell proliferation, inducing metabolic alterations in energy consumption, and activating M1-like macrophages. By implementing an Sptlc2-macrophage deficient model, the authors investigated the role of sphinganine in LPS-induced endotoxemia and tumor control. However, there are many caveats with this manuscript, including a lack of appropriate control for critical experiments and inappropriate conclusions that dampened the enthusiasm for this manuscript.

Major comments:

- 1) There is an overall lack of quantification of the immunoblots in every experiment and statistical analysis in the figures.
- 2) The way the authors described the animal models is confusing. Usually, Lyz2-cre is considered WT. This reviewer got confused multiple times in the manuscript. If LyZ-cre also expresses Sptlc2, this strain should be Sptlc2^{fl/fl}_lyz2-cre. Also, what is the actual WT strain used here? Furthermore, the deletion efficiency is not apparent.
- 3) Figure 2 is problematic since most data are speculative. The pH of the supernatant should be further confirmed using a different technique (not the pH strips). Figure 2SA: The authors use a qualitative evaluation of the pH alteration in the cell culture medium; however, without the indicative reference staining, it is impossible to follow the indicated conclusions. Also, the confluence in part B is somewhat confusing since BMDMs should not proliferate upon LPS stimulation. If you compare PBS vs. LPS, it is clear that LPS increases cell proliferation. How is this happening? Fig. 2C, showing single cells without any quantification, is relatively weak. Therefore, the authors should quantify the size and morphology of the cells in all experimental groups.
- 4) In Fig. 3B, the authors suggest that Sptlc2 regulates MyD88 expression. However, there is no housekeeping protein, along with appropriate quantification, to determine the relative expression of these proteins. In Fig 3C, MFI, in addition to %, should be shown. In Fig. 3D and F, it is surprising the MyD88 seems to be localized in the nucleus in both resting and LPS-challenged cells. Nonetheless, there is only one cell shown/group and no quantification of colocalization. Figure 3D, it is possible to see differences in the fluorescence merged after LPS stimulation; however, is this an actual decrease in recruitment of MyD88, or is it a consequence of less expression of MyD88 (Figure 3C)?
1. Fig 4 is interesting; however, Fig 4 B does not show binding to MyD88. Also, panel D seems overexposed, and it is not clear what the authors are trying to show. I suggest another way to confirm this colocalization, such as FRET, FRAP, or PLA—the same suggestion for colocalization with TIRAP.
- 5)
- 6) Line 208: 'Sptlc2-deficient macrophages significantly decreased in size after LPS stimulation'. In this phrase, it is unclear whether the author means a decrease in the population percentage after stimulation or a reduction in the cellular size. Dot plots are not the best way to show the size of cells; in this case, the authors must use a complementary approach to confirm these results.
- 7) In Fig 5, the authors state that LPS is a sepsis model. However, LPS is only an inert particle and does not represent an active infection. This is a model of endotoxemia. If the authors want to claim that sptlc2 is relevant for sepsis, the authors should perform a model of cecum ligation and puncture or i.v. infection with relevant gram-negative bacteria.

8) Figure 5E-F lacks data from control naïve animals. This is because it is unclear whether the changes in the manuscript are related to the stimulation or whether there are already differences in naïve cells.

9) Fig 6 seems tangential to the main findings, and the LPS detection seems exploratory and irrelevant to the main results.

Minor comments:

1) In Figure 1, it is unclear why the authors performed macrophage polarization experiments using LPS +IFN and then, in further experiments, only LPS.

REVIEWER COMMENTS

Reviewer #1 (Remarks to the Author):

Reviewer comments to the MS entitled 'Sphinganine membrane-anchors TLR4 adaptors in macrophages to promote inflammation'. Submitted to Nat. Comm. NCOMMS-23-28052-T by Hering M, Madi A, Sandhoff R, Ma S, Wu J, Mief A, Richter K, Mohr K, ten Bosch N, Stichling D, Poschet G, Umansky V and Cui G. The MS is presenting impressive results and well written, covering biological phenomena up to elaborating molecular mechanisms to unravel a novel role of the sphingolipid sphinganine, which was previously held to function as a building block for other sphingolipids, especially sphingolipid-1-phosphate and ceramides, at the interfaces between pattern recognition as a function of the innate immune system and subsequent host response on the level of gene expression and release of proinflammatory cytokines. Experiments were well performed and described, statistical were adequately performed. Despite the impressive and interesting findings, there are some major and minor points of criticism prior to debate the acceptance of the MS to improve the quality of that.

Reply: We thank Reviewer #1 for the positive comments such as "The MS is presenting impressive results and well written" and "[...] it unravels a novel role of the sphingolipid sphinganine". We also thank Reviewer #1 for other comments that improve our manuscript.

Major points:

- Administration of LPS is not a sepsis model (Buras JA et al 2005). To improve the quality of the MS and to expand the outreach of your impressive results for clinical research, you have to perform a comparison of *Sptlc2*-deficient mice with wt in an established model of polymicrobial sepsis.

Reply: We thank Reviewer #1 for raising this very important point. Because Reviewer #2 and Reviewer #3 asked similar questions, we will address these questions together. As pointed out by the reviewers, the LPS model is a mouse model for studying endotoxemia. To investigate the role of *Sptlc2* in polymicrobial sepsis, we used the cecal slurry (CS) model (**Reviewer Figure 1**). This model has been widely used to study sepsis¹⁻³. It has several advantages over other polymicrobial models: First, as we are interested in the very-early, highly pro-inflammatory phase of the sepsis, in comparison to the CLP model, which requires surgery of the mice and recovery of the surgery, the CS model is better suited. Second, in the CLP model different cecum shapes and sizes increase the variability, while in the CS model the injected dose of CS can be accurately determined. Third, as differences in the microbiome of the *Lyz2*-cre mice cannot be entirely excluded, it is reasonable to induce the sepsis by the same inducing agent in all mice. Our findings demonstrate that *Sptlc2* played similar roles in CS-induced sepsis model and the LPS-induced endotoxemia model. For example, similar to the LPS-induced endotoxemia model, *Sptlc2* levels were increased in intraperitoneal macrophages during CS-induced sepsis (**Reviewer Figure 1B**). Intraperitoneal macrophages lacking *Sptlc2* expressed lower levels of M1-like marker CD38 and higher levels of M2-like marker Arg-1 (**Reviewer Figure 1C**). Cytokine array analysis revealed a trend towards reduced levels of most of the measured cytokines, chemokines and acute-phase proteins in *Lyz2*-cre mice (**Reviewer Figure 1D**). However, unlike in the LPS-induced endotoxemia model, most of the cytokine measurements did not reach statistical significance between WT and *Lyz2*-cre mice. One possible explanation is that not only LPS, but also other bacterial stimuli induced cytokine production in the CS-induced polymicrobial sepsis model. *Sptlc2* may play a more important role in the LPS/TLR4 signaling pathway than in other innate pattern recognition receptor-mediated pathways. Therefore, *Sptlc2* deficiency caused a stronger

phenotype in the LPS-induced endotoxemia model than in the CS-induced polymicrobial sepsis model. We also agree with the three reviewers that the term “LPS-induced sepsis” (or “endotoxemia”) should be used instead of only the broad term “sepsis”. We have adapted the manuscript accordingly. Furthermore, we totally agree that models of polymicrobial sepsis add important value to our findings and we are very grateful for the comment. Therefore, we included the experimental results of the CS-induced polymicrobial sepsis model (Reviewer Figure 1) in the revised manuscript as **Supplementary Figure S6**.

Reviewer Figure 1 Deficiency in *Sptlc2* alleviates pro-inflammatory M1-like macrophage phenotype and cytokine production in cecal slurry-induced polymicrobial sepsis

(A) Illustration of the experimental design of the polymicrobial sepsis model. Cecal slurry was intraperitoneally injected into WT or *Ly2z2-cre* mice. 3 hours post injection, mice were sacrificed, and samples for cytokine arrays and flow cytometry analyses were isolated.

(B) Bar graph, showing MFI of *Sptlc2* in WT or *Ly2z2-cre* intraperitoneal macrophages 3 hours post injection, measured by flow cytometry (PBS: N=3, Cecal slurry: N=6).

(C) Bar graph, showing percentages of CD38+ or Arg-1+ macrophages collected by intraperitoneal flushing 3 hours after cecal slurry injection (N=6).

(D) Bar graph, showing the pixel density of cytokine dots in *Lyz2-cre* plasma normalized to averaged WT plasma. Quantification was performed in FIJI. A black dashed line separates up- and down-regulated cytokines in *Lyz2-cre* plasma. Cytokine array blots spotted in duplicates with capture antibodies to specific target proteins were probed with plasma isolated from WT or *Lyz2-cre* mice 3 hours after PBS (N=3) or LPS injection (N=6). For plasma of each mouse a new membrane was used, and the layout of the array can be found in **Supplementary Figure S5G**. The data are presented as mean \pm SD (B-C) or mean + SEM (D). Statistical comparisons were performed with ANOVA tests (B; for simultaneous comparisons of more than two groups), two-tailed Student's *t* tests (C-D; data points were normally distributed).

- the role of cholesterol and ceramides in your experimental setting is unclear. In both conditions, infection and (sterile) inflammation, these two compounds are critically involved in assembly of membrane embedded proteins/receptors, relevant for PAMP-triggered signal transduction.

Reply: Thank you for this comment. In agreement with literature, we found that ceramides were increased after M1-like activation (**Main Figure 1B, E**), underlining your point that they might have crucial roles in macrophages during infection and inflammation. We also found ceramides to be decreased upon *Sptlc2* deficiency (**Main Figure 2F**). Following Reviewer #1's comments, we investigated the potential effects of cholesterol and ceramides, together with sphinganine in our experimental setting. Similar as shown in **Main Figure 2G**, we found that sphinganine increased cell sizes of *Sptlc2*-deficient cells to a level comparable of *Sptlc2*-sufficient cells (**Reviewer Figure 2**). Neither cholesterol nor ceramides rescued the sizes of *Sptlc2*-deficient BMDM (**Reviewer Figure 2**). Although ceramide and cholesterol did not rescue BMDM sizes of *Sptlc2*-deficient cells, they showed a trend to increase BMDM size. This is in line with Reviewer #1's comments that cholesterol and ceramide may regulate the assembly of membrane embedded proteins and further influence cell sizes. Sphinganine is the precursor molecule to synthesize ceramides. The observation that sphinganine but not ceramides increased cell size of *Sptlc2*-deficient cells suggests that sphinganine does not need to be converted to ceramide to increase the cell size of *Sptlc2*-deficient cells. **Reviewer Figure 2** is included in the revised manuscript as part of **Supplementary Figure S2E**.

Reviewer Figure 2 Other than cholesterol, ceramide and sphingosine-1-phosphate, sphinganine completely restores size of LPS-activated BMDM

Bar graph, showing percentages of FSC-A^{high}, SSC-A^{high} WT or *Lyz2-cre* LPS-activated BMDM after DMSO, sphinganine (Sa), cholesterol (Chol), sphingosine-1-phosphate (S1P) or ceramide supplementation for 3 days (N=3). The data are presented as mean ± SD. Statistical comparisons were performed with ANOVA tests (for simultaneous comparisons of more than two groups).

- sphingosine-1-phosphate is the most important product of metabolism of sphingosine and essential for a broad range of immune response mechanisms (membrane stabilization etc. etc.) without providing data on S1P concentrations. There is a lack of ining your arguments.

Reply: We want to thank Reviewer #1 for raising this point. Following this comment, we determined the concentrations of S1P. We found that *Sptlc2* deficiency reduced the concentrations of S1P from 21.9 ± 7.7 pmol/million cells to 7.8 ± 2.8 pmol/million cells (**Main Figure 2F**). However, supplementing S1P did not increase the cell sizes of *Sptlc2*-deficient cells (**Reviewer Figure 2**). These results suggest that although the *Sptlc2* deficiency reduced the abundance of S1P, S1P is not primarily responsible for the reduced sizes of BMDM. **Reviewer Figure 2** is included in the revised manuscript as part of **Supplementary Figure S2E**.

Minor points:

- Nomenclature of endotoxin subtypes throughout the MS with an alphabetical character 'O' and not a digit 'zero'.

Reply: Thank you for this comment; we have adapted accordingly.

- in graphical abstract the term 'sepsis symptoms' is misleading, 'immune response' or anything else is more appropriate.

Reply: We totally agree with this suggestion. Following this suggestion, we have rephrased it to "acute inflammatory response".

- the alphabetical characters N and C determining the terminus of a protein must be given in italic typeface.

Reply: That is a very good point. Thank you. We have adapted accordingly (*N*-myristoylation, *N*-acyl, *C*-terminus, *N*-terminus etc.).

Once again, we would like to emphasize our gratitude for Reviewer #1's valuable input. These comments were all very helpful and we highly appreciate them.

Reviewer #2 (Remarks to the Author):

This is a fairly complex paper about the role of sphingolipids in TLR4 mediated signaling. In particular, the authors found that LPS induces the production of various sphingolipids, particularly sphinganine, which helps recruit MyD88 to the cell membrane to form a complex with TIRAP and the cytoplasmic portion of TLR4. They generated a conditional KO of *Sptlc2*, a key component of the sphingolipid biosynthetic pathway. Mutant cells deficient in *Sptlc2* had a much diminished inflammatory response to LPS compared with *Sptlc2* expressing cells, and the authors present evidence that this may be due to decreased MyD88 recruitment. Overall, *Sptlc2* deficiency decreased inflammatory responses to injected LPS. Finally, *Sptlc2* deficiency reduced the activity of macrophages against a melanoma tumor line in vivo, which they posited might be due to decreased responses to LPS within the tumor. While I learned a lot reading this paper and feel it makes a nice contribution to our knowledge about sphingolipid biosynthesis in LPS responses, I was struck by a few things. First, the paper is difficult to read. Many of the sentences are very long and complex, when simpler language would suffice. This may be due to the fact that the authors are not native English speakers, and is not meant to be a criticism. However, the paper would be improved with editing.

Reply: Thank you for the positive feedback. We shortened long sentences wherever possible to improve the language and also did some further editing.

Second, I think the authors sometimes overreach in drawing conclusions. An example of this is the experiment where they express MyrMyD88 in mutant myeloid cells. MyrMyD88 did, in fact, alter the phenotype of *Sptlc2*-deficient macrophages towards that of wild type macrophages, but this could be explained in several ways and they chose only one explanation for the data.

Reply: We really appreciate Reviewer #2's comments. To more stringently examine the role of the membrane-anchored MyrMyD88 in *Sptlc2*-deficient macrophages, we have repeated the experiment with two additional controls: a wild-type (WT) form of MyD88 and also a mutated version of MyD88 (**Reviewer Figure 3**). The mutation was introduced at position L252P by replacing "CTG" (encoding Leucine) with "CCG" (encoding Proline) by site-directed mutagenesis (**Reviewer Figure 3A**). This mutation in the TIR domain of MyD88 causes MyD88 oligomerization but did not influence its membrane location⁴. Overexpression of WT or mutated MyD88^{L252P} in BMDM did not significantly rescue the cell size of *Sptlc2*-deficient macrophages (**Reviewer Figure 3B**). Therefore, with the newly added data, we conclude that neither the amount of MyD88 protein nor the oligomerization of MyD88 protein restores the sizes of *Sptlc2*-deficient macrophages completely. Instead, it is the membrane location that increases the size of *Sptlc2*-deficient macrophages. We hope these additional controls help to clarify Reviewer #2's concerns over this issue. Furthermore, we also agree with Reviewer #2 and we do not exclude alternative mechanisms of action through which sphingolipids recruit adaptor proteins besides MyD88. **Reviewer Figure 3** is included in the revised manuscript as part of **Supplementary Figure S3**.

Reviewer Figure 3 Overexpression of WT or L252P-mutated MyD88 did not completely rescue BMDM size after LPS activation

(A) Illustration of the experimental design in B. WT or L252P-mutated MyD88 were overexpressed in WT or *Lyz2-cre* BMDM with X-treme GENETM HP transfection of HEK293T cells with MigR1-GFP-MyrisMyD88 and the eco helper plasmid, and subsequent retroviral transduction with viral particles of WT or *Lyz2-cre* BMDM.

(B) Bar graph, showing percentages of FSC-A^{high}, SSC-A^{high} GFP+ LPS-stimulated BMDM (N=3). Data are presented as mean ± SD (B). Statistical comparisons were performed with ANOVA tests (B; for simultaneous comparisons of more than two groups).

I was particularly surprised by their LPS "sepsis" model: rather than perform the studies as is usually done (ie, to inject increasing doses of LPS to induce partial lethality), they measure hunching and loss of movement. These are difficult criteria to assess. I assume that the reason they did not examine mortality was related to an ethical concern, but the authors would benefit by stating this if it is true.

Reply: We want to thank Reviewer #2 for raising this point. As mentioned by Reviewer #2, due to ethical reasons that "with animal models, death as an intentional end point is ethically unacceptable"⁵, mortality measurement was not possible due to the high animal suffering and the ethical concerns. We chose these parameters based on literature that describe comparable sepsis scoring systems^{5,6}. While we experienced that accurate measurement of e.g. eye opening and

respiration rate are quite difficult to assess, we realized that loss of movement and a hunched posture of the mice, were very clear symptoms of the LPS-induced sepsis, which were also used before in literature to assess sepsis severity^{5,6}. These signs also marked the endpoint criteria in these experiments and can also be seen in full movement in the provided video, suggesting that these are useful parameters to assess LPS-induced sepsis severity.

Measuring cytokine and other inflammatory markers is certainly an acceptable alternative to lethal LPS experiments, but in this regard, the differences between cytokines in the KO vs wild-type macs was not impressive, even if significant statistically. I suggest that these experiments would benefit from a time course and/or a dose response experiment. The authors should recognize that 6 hours may be rather late to be measuring cytokines. For example, TNF peaks about 1.5 hours after LPS challenge and would certainly have been worth measuring.

Reply: Thank you for this advice. The cytokine measurements were not performed 6 hours after injection as stated in the reviewer's comment, but 3 hours after the injection (please find this information also in the legend of **Main Figure 5G, H**). As suggested by the reviewer, we had performed a time-course analysis of cytokines after LPS injection (**Reviewer Figure 4**). We found that cytokines can be classified in 3 groups: the early-induced (**Reviewer Figure 4A**), medium-induced (**Reviewer Figure 4B**) and late-induced cytokines (**Reviewer Figure 4C**). Based on this, we decided to use 3 hours post injection as ideal time-point for blood collection, as the most relevant cytokines induced through TLR4 stimulation were measurable in the plasma during this time. We completely agree with the reviewer that e.g. TNF α peaks very early after injection and this is also represented in our measurements (**Reviewer Figure 4A**). Moreover, we found our kinetics to be in line in large parts with previous reports from literature measuring LPS-induced cytokines with a different method⁷. **Reviewer Figure 4** is included in the revised manuscript as part of **Supplementary Figure S5**.

Reviewer Figure 4 Plasma cytokine kinetics in WT mice post PBS or LPS injection

(A) Time-course of levels of the early-induced cytokines TNF α and IL-10 in the plasma of WT mice (N=1-2).

(B) Time-course of levels of the medium-induced cytokines IL-1 β , IL-6 and IL-18 in the plasma of WT mice (N=1-2).

(C) Time-course of levels of the late-induced cytokines IL-12/IL-23 p40, IL12 p70 and IFN γ in the plasma of WT mice (N=1-2). Data are presented as mean \pm SD (A-C).

Finally, LPS injection does not equal "sepsis" (this view of LPS challenge was discarded many years ago), and should not be referred to as such.

Reply: We thank Reviewer #2 for raising this very important point. Because Reviewer #1 and Reviewer #3 asked similar questions, we have followed the reviewers' suggestions and rephrase it as "LPS-induced sepsis model" (or "endotoximea"). In addition, we have followed the reviewers' comments and repeated the experiments using a polymicrobial sepsis model. The results are shown in **Reviewer Figure 1** and **Supplementary Figure S6**.

While I find the anti tumor experiments to be interesting, I think the authors might consider removing them from this manuscript and focusing on LPS signaling. The experiments on the anti tumor effect should be enlarged upon, and could easily be part of a followup manuscript.

Reply: Thank you for this comment. We have discussed with the editorial office. It is believed that the experimental results of the tumor model provide further insights into the role of Sptlc2 in

myeloid cells in inflammation. Therefore, following this suggestion, we will keep these results in the current manuscript.

We thank Reviewer #2 very much for the valuable input. The made modifications based on these comments consolidate our observations and make our manuscript easier to understand.

Reviewer #3 (Remarks to the Author):

Hering et al. sought to investigate the role of bioactive lipid sphinganine in mediating toll-like receptor activation in macrophages. The authors employed a variety of approaches, along with M1 and M2 macrophage differentiation, lipidomic analysis, and different pulldown experiments to demonstrate that sphinganine is essential for the expression of the TIR adaptor MyD88 and its partner TIRAP, as well as their recruitment to the cell membrane and subsequent activation of the NF- κ B pathway upon LPS activation. Additionally, they observed that LPS stimulation induces the expression of sphinganine, promoting cell proliferation, inducing metabolic alterations in energy consumption, and activating M1-like macrophages. By implementing an Sptlc2-macrophage deficient model, the authors investigated the role of sphinganine in LPS-induced endotoxemia and tumor control. However, there are many caveats with this manuscript, including a lack of appropriate control for critical experiments and inappropriate conclusions that dampened the enthusiasm for this manuscript.

Major comments:

1) There is an overall lack of quantification of the immunoblots in every experiment and statistical analysis in the figures.

Reply: Thank you for your comment. The immunoblot in **Main Figure 1F** contains a quantification (numbers indicate the relative intensity of Sptlc2 signal normalized to the loading control). It is correct, the immunoblots in **Main Figures 3B** and **6A** do not contain quantifications yet. We did not add them initially, as in **Main Figure 3B**, the loading controls (Actin and Grp94) show a pretty even gel loading and the differences in the levels of the measured proteins are quite obvious visually and in parts even further confirmed more quantitatively by flow cytometry or by confocal fluorescence microscopy. Following Reviewer #3's request, we now quantified the immunoblot shown in **Main Figure 3B (Reviewer Figure 5)**. In **Main Figure 6A**, we quantified LPS O55:B5 using the LAL assay.

Reviewer Figure 5 Immunoblot including quantification from Main Figure 3B

Band intensities were normalized to the intensities of the loading controls Grp94 and Actin and quantification of band intensities are presented as numbers above the respective bands.

2) The way the authors described the animal models is confusing. Usually, Lyz2-cre is considered WT. This reviewer got confused multiple times in the manuscript. If LyZ-cre also expresses Sptlc2, this strain should be *Sptlc2^{fl/fl}_lyz2-cre*. Also, what is the actual WT strain used here? Furthermore, the deletion efficiency is not apparent.

Reply: Thank you for this comment. To clarify, we refer to *Sptlc2^{Flox/Flox} Lyz2-cre* positive mice as “Lyz2-cre mice”, and we refer to *Sptlc2^{Flox/Flox} Lyz2-cre* negative mice as “WT” mice. We have defined these abbreviations the first time we used them in the main text (first section of results of the revised manuscript). In more detail, in all mice exon 1 of *Sptlc2* is flanked by two *LoxP* sites and is excised after crossing with a Cre-expression mouse strain. Therefore, mice that express a *Lyz2-cre* are *Sptlc2*-deficient, while mice that do not express a *Lyz2-cre* are not. These mice which are not deficient for *Sptlc2* are referred to as WT. The deletion efficiency is presented for each of the different models: the BMDM model in **Main Figure 1F**, the LPS-induced sepsis model in **Main Figure 1G** and the melanoma model in **Main Figure 6F**. We also provided these information in the corresponding method part.

3) Figure 2 is problematic since most data are speculative. The pH of the supernatant should be further confirmed using a different technique (not the pH strips). Figure 2SA: The authors use a qualitative evaluation of the pH alteration in the cell culture medium; however, without the indicative reference staining, it is impossible to follow the indicated conclusions. Also, the confluence in part B is somewhat confusing since BMDMs should not proliferate upon LPS stimulation. If you compare PBS vs. LPS, it is clear that LPS increases cell proliferation. How is this happening? Fig. 2C, showing single cells without any quantification, is relatively weak. Therefore, the authors should quantify the size and morphology of the cells in all experimental groups.

Reply: Thank you for your comments. To clarify, the pH strips were the additional confirmation of the pH measurement that was conducted with a pH meter (pH 50 Benchmeter, VioLab). For more information please see the corresponding material and method part. The pH measurement quantification in **Main Figure 2A** is not based on the pH strips. We agree the pH strips would need a legend to be fully comprehensive. We believe the pH meter measurement is more accurate and therefore, based on these comments, we now believe it might be less misleading if we exclude the pH strips from the figure set and focus only on the pH meter measurements. We agree with the comment in that we do not believe that primary BM-derived macrophages proliferate in cell culture, but we want to emphasize that we did not describe this. Confluency measurements do not necessarily measure cell proliferation but rather the area of the plate covered by the cells. There can be several reasons for that: One of these could be proliferation, another could be increased cell death. We did not see changes in cell death between *Sptlc2*-sufficient and *Sptlc2*-deficient BMDM. Another reason could be cell growth of the already existing cells. Therefore, rather, our confluency measurements show that the existing BMDM grow in size. This phenomena is stronger in the WT than in the *Lyz2*-cre BMDM and therefore results in increased confluency in the WT group. This was the reason we had a closer look on the detailed morphology of the BMDMs by scanning electron microscopy (**Main Figure 2C**) and also assessed cell size by flow cytometry (**Main Figure 2G**) and confocal fluorescence microscopy (**Main Figure 3D, F**) as a parameter of M1-activation. In line with literature ^{8,9} suggesting that LPS-stimulation increases size of BMDM and our confocal fluorescence microscopy images, our data further indicate that this phenomena is affected by *Sptlc2* deficiency and can be rescued through restoring MyD88 membrane recruitment. We agree with the comment that in scanning electron microscopy pictures different parameters can be assessed and we tried to name the most obvious changes. Still, we believe that the scanning electron microscopy pictures are of high use for the reader to get a better visual idea of the changed morphology of the BMDM after LPS-activation and *Sptlc2* deletion, but for quantitative measurements we believe flow cytometric size measurements or quantifications from the confocal fluorescence microscopy images, in which the cell borders can be determined more accurately and cell numbers are higher, allowing for better statistical comparisons, are better suited.

4) In Fig. 3B, the authors suggest that *Sptlc2* regulates MyD88 expression. However, there is no housekeeping protein, along with appropriate quantification, to determine the relative expression of these proteins. In Fig 3C, MFI, in addition to %, should be shown. In Fig. 3D and F, it is surprising the MyD88 seems to be localized in the nucleus in both resting and LPS-challenged cells. Nonetheless, there is only one cell shown/group and no quantification of colocalization. Figure 3D, it is possible to see differences in the fluorescence merged after LPS stimulation; however, is this an actual decrease in recruitment of MyD88, or is it a consequence of less expression of MyD88 (Figure 3C)?

Reply: Thank you for this comment. Actually, there are housekeeping controls and in the legend of **Main Figure 3B** it is written that Actin and Grp94 were used as loading controls. After adjusting the protein amount to be loaded by BCA assay in each of our immunoblotting experiments, these proteins ensure an equal gel loading, while MyD88, p65 (Ser536) and I κ B α levels are altered by *Sptlc2* deficiency (**Main Figure 3B**). For optimal quantitative comparisons we then used flow cytometry (**Main Figure 3C**). Following the Reviewer's comment, we have also analyzed the MFI of MyD88 in BMDM with and without LPS activation and the results validate the ones showing percentages of MyD88+ macrophages in **Main Figure 3C** (**Reviewer Figure 6**).

Reviewer Figure 6 *Sptlc2* is required for the LPS-induced upregulation in MyD88 levels

Bar graph, showing the MFI of MyD88 in WT or *Lyz2-cre* BMDM after PBS or LPS stimulation for 20 hours. Data are presented as mean \pm SD. Statistical comparisons were performed with ANOVA tests (for simultaneous comparisons of more than two groups).

In **Main Figures 3D, F**, we agree there is a MyD88 signal in the nuclear region for some cells. Even though we did not further discuss this observation, we want to mention that nuclear MyD88 has been described before¹⁰. We provided images including more cells in the supplementary (**Main Figure S3B**). In these, white arrows highlight several cells, for which we observed the described phenotype and of which representative cells were picked and shown enlarged in the main figure set. For drawing the attention to the subcellular localization of MyD88 and TLR4, we believe it's better to focus on a single cell in the main figure set. The membrane-anchoring of MyD88 can be visually seen in these representative cells. Regarding the last part of the comment, we believe in the *Lyz2-cre* BMDM it is both: a decrease in recruitment of MyD88 and less expression of MyD88. For more details please see our response to Reviewer #2 and the WT and mutated MyD88 overexpression (**Reviewer Figure 3**).

5) Fig 4 is interesting; however, Fig 4B does not show binding to MyD88. Also, panel D seems overexposed, and it is not clear what the authors are trying to show. I suggest another way to confirm this colocalization, such as FRET, FRAP, or PLA—the same suggestion for colocalization with TIRAP.

Reply: Thank you so much for this comment. We agree the data were not presented in the most comprehensive way, and we have followed your advice to improve it. We now only show the relevant part of the Coomassie Blue-stained gel, from which a small gel block was cut out (as indicated by the box) and analyzed by LC-MS/MS (**Reviewer Figure 7**). Marked by stars, we now

highlight in which of the lanes MyD88 was identified in the LC-MS/MS experiment. We used the same way to present the LC-MS/MS data previously ¹¹. In more detail, it is highlighted now in the figure that we could only identify MyD88 after Sa-biotin, but not control-biotin pulldown. We then further validated these findings by antibody-dependent staining in **Main Figure 4C**. Additionally, we show the raw data of the LC-MS/MS in the **Supplementary Table S2**. For better understanding in **Reviewer Figure 7B**, we show the LC-MS/MS output, validating the presence of MyD88 only after Sa-biotin, but not control-biotin pulldown. **Reviewer Figure 7A** is included in the revised manuscript as part of **Main Figure 4**.

Reviewer Figure 7 SpHINGANINE-biotin pulldown identified physical interaction of sphinganine and MyD88

(A-B) Proteins in the indicated Coomassie gel regions (dotted boxes in A) after sphinganine (Sa)-biotin or control biotin pulldown were identified by LC-MS/MS (B). MyD88 was only identified after Sa-biotin, but not control biotin pulldown. Additional information is provided in **Supplementary Table S2**.

Regarding panel D, based on the reviewer's comment, we believe it is better to focus on only one condition, which in this case is the PBS-treated one. Here, we wanted to show that also from a spatial point of view, an interaction of sphinganine and adaptor proteins such as TIRAP is likely, as they co-localize within the cell (**Reviewer Figure 8**). This co-localization can clearly be seen, as the co-localization of the green and red stained molecules results in a yellowish signal in the merged picture (**Reviewer Figure 8**). As we showed that LPS induces Sptlc2 activity, LPS induces endogenous sphinganine production which in this experiment would compete with the supplemented fluorescently-labeled sphinganine. However, TIRAP expression levels were shown

before not to be influenced by LPS stimulation (**Main Figure 3B**). Therefore, we believe the PBS-treated samples are better suited in this experiment. **Reviewer Figure 8** is included in the revised manuscript as part of **Main Figure 4**.

Reviewer Figure 8 Supplemented sphinganine co-localizes with TIRAP in BMDM

Confocal fluorescence microscopy images, showing subcellular localization of supplemented sphinganine (Sa)-fluorescein and TIRAP in WT BMDM after PBS treatment. Data are representative of 12 images from four mice.

To confirm the co-localization, we think in our study FRAP is not suited, as this method only works in cells without fixation. However, to stain for intracellular MyD88 with primary and secondary antibodies we have to fix the cells in our experiments. While we appreciate the utility of proximity ligation assays, we find it not possible to use this technique for our purposes as no antibody against sphinganine is available. FRET can be useful to assess interactions of molecules in close proximity. However, FRET can only be detected if the distance between the analyzed molecules is less than ~10 nm. Given that we have to use a primary (anti-MyD88 or anti-TIRAP) and secondary (anti-rabbit) antibody to detect the adaptors, the detection of FRET is also highly dependent the relative orientation of the donor emission dipole moment and the acceptor absorption dipole moment. Still, following Reviewer #3's comment, we used this method (in more detail: Acceptor photo-bleaching (AP) – FRET analysis) and we could detect FRET especially in the membrane area (ROI2) of sphinganine-fluorescein-supplemented and MyD88-TRITC stained BMDM (**Reviewer Figure 9**). Since the donor and acceptor are co-localized only in certain regions, we propose to include these images in the supplementary section of the revised manuscript. **Reviewer Figure 9** is included in the revised manuscript as part of **Supplementary Figure S4**.

Reviewer Figure 9 Acceptor photo-bleaching (AP) – FRET analysis

After supplementing sphinganine-fluorescein (AP-FRET donor) to the BMDM for 3 days and staining for MyD88 (rb) + anti-rb-TRITC (AP-FRET acceptor), AP-FRET within the photo-bleached region of interest (ROI1), especially close to the membrane (ROI2), could be detected, while no AP-FRET was detectable in the non-photo-bleached control regions of interest (ROI3, ROI4), as expected. Data are representative of 3 images.

6) Line 208: 'Sptlc2-deficient macrophages significantly decreased in size after LPS stimulation'. In this phrase, it is unclear whether the author means a decrease in the population percentage after stimulation or a reduction in the cellular size. Dot plots are not the best way to show the size of cells; in this case, the authors must use a complementary approach to confirm these results.

Reply: Thank you for this comment. We agree with the reviewer and we adapted the text accordingly. It is more accurate to write “the percentage of large (FSC-A^{high}, SSC-A^{high}) BMDMs was significantly reduced upon *Sptlc2* deletion after LPS stimulation”. Size differences between WT and *Ly2z2*-cre mice and after PBS or LPS stimulation become obvious in the confocal fluorescence measurement in **Main Figure 3D, F** and the corresponding size quantification. Similar macrophage size differences have also been reported before between PBS and LPS-activation^{8,9}.

7) In Fig 5, the authors state that LPS is a sepsis model. However, LPS is only an inert particle and does not represent an active infection. This is a model of endotoxemia. If the authors want to claim that *sptlc2* is relevant for sepsis, the authors should perform a model of cecum ligation and puncture or i.v. infection with relevant gram-negative bacteria.

Reply: Thank you. Because Reviewer #1 and Reviewer #2 asked similar questions, we have followed the reviewers' suggestions and rephrase it as “LPS-induced sepsis model” (or “endotoxemia”). In addition, we have followed the reviewers' comments and repeated the experiments using a polymicrobial sepsis model. The results are shown in **Reviewer Figure 1** and **Supplementary Figure S6**.

8) Figure 5E-F lacks data from control naive animals. This is because it is unclear whether the

changes in the manuscript are related to the stimulation or whether there are already differences in naïve cells.

Reply: Thank you for your comment. Actually, we did include PBS-injected mice in these experiments. Please find them in **Reviewer Figure 10**. Similar to our *in vitro* BMDM findings, we found that only upon LPS stimulation, WT and *Lyz2-cre* macrophages showed phenotypic and numerical differences. We did not observe significant differences in macrophage Arginase-1 levels or macrophage numbers in the intraperitoneum of PBS-injected WT and *Lyz2-cre* mice (**Reviewer Figure 10**). Thereof, we conclude the role of sphingolipids in LPS-induced TLR4 signaling is primarily visible upon activation of this pathway and not during steady-state signaling during macrophage differentiation without stimulus. **Reviewer Figure 10** is included in the revised manuscript as part of **Supplementary Figure S5**.

Reviewer Figure 10 Macrophage numbers and Arg-1 expression are unchanged by *Sptlc2* deficiency in PBS-injected mice.

(A) Bar graph, showing fold change of WT or *Lyz2-cre* intraperitoneal macrophages 6 hours after PBS injection, measured by flow cytometry (N=3).

(B) Bar graph, showing fold change of Arg-1+ WT or *Lyz2-cre* macrophages collected by intraperitoneal flushing 6 hours after PBS injection (N=3). The data are presented as mean \pm SD (A-B). Statistical comparisons were performed with two-tailed Student's t tests (A-B; data points were normally distributed).

9) Fig 6 seems tangential to the main findings, and the LPS detection seems exploratory and irrelevant to the main results.

Reply: Thank you for raising this point. Because also Reviewer #2 asked this question, we have discussed with the editorial office. It is believed that the experimental results of the tumor model provide further insights into the role of *Sptlc2* in inflammation. Therefore, following this suggestion, we will keep these results in the current manuscript.

Minor comments:

1) In Figure 1, it is unclear why the authors performed macrophage polarization experiments using LPS +IFN and then, in further experiments, only LPS.

Reply: Thank you very much for this comment. We have now adapted the respective section to clarify why we started with the double stimulus and then went on with “only LPS” (from **Main**

Figure 1F on). We initially followed the literature and use LPS+IFN γ as M1-like stimulus. We found this induces certain sphingolipids including sphinganine (**Main Figure 1A**). Re-analyzing the RNA data in **Main Figure 1C** showed to us that LPS alone is enough to induce Sptlc2. We validated this on protein level in **Main Figure 1E**, by showing that the combinatorial stimulus induces Sptlc2 and its product ceramide. To be able to narrow down on 1 pathway instead of the combination, in line with literature, we then proved that also LPS alone induces Sptlc2 also on protein level (**Main Figure 1F**)²⁰. From here on, we specifically looked at LPS induced pathways and thereby we could identify the sphingolipid-driven regulation of TLR4 signaling specifically. Notably, in **Main Figure 5I**, we again used the double stimulus, which is because LPS alone is not enough to induce full expression of certain cytokines (e.g. IL-12 p70) in *in vitro* differentiated BMDMs. This has been reported multiple times in literature²¹⁻²³. In the *in vivo* cytokine measurements, IFN γ is provided by e.g. intraperitoneal ILC's, NK cells and unconventional T cell subsets. We thank Reviewer #3 very much for the comments. We believe that by including the inquired controls our manuscript is easier to follow and more convincing.

References

- 1 Fanti, A. K. *et al.* Flt3- and Tie2-Cre tracing identifies regeneration in sepsis from multipotent progenitors but not hematopoietic stem cells. *Cell Stem Cell* **30**, 207-218 e207, doi:10.1016/j.stem.2022.12.014 (2023).
- 2 Starr, M. E. *et al.* A New Cecal Slurry Preparation Protocol with Improved Long-Term Reproducibility for Animal Models of Sepsis. *PLOS ONE* **9**, e115705, doi:10.1371/journal.pone.0115705 (2014).
- 3 Cai, L., Rodgers, E., Schoenmann, N. & Raju, R. P. Advances in Rodent Experimental Models of Sepsis. *International Journal of Molecular Sciences* **24** (2023).
- 4 Schmidt, K. *et al.* B-Cell-Specific Myd88 L252P Expression Causes a Premalignant Gammopathy Resembling IgM MGUS. *Front Immunol* **11**, 602868, doi:10.3389/fimmu.2020.602868 (2020).
- 5 Huet, O. *et al.* Ensuring animal welfare while meeting scientific aims using a murine pneumonia model of septic shock. *Shock* **39**, 488-494, doi:10.1097/SHK.0b013e3182939831 (2013).
- 6 Shrum, B. *et al.* A robust scoring system to evaluate sepsis severity in an animal model. *BMC Res Notes* **7**, 233, doi:10.1186/1756-0500-7-233 (2014).
- 7 Liu, J. *et al.* Screening cytokine/chemokine profiles in serum and organs from an endotoxic shock mouse model by LiquiChip. *Sci China Life Sci* **60**, 1242-1250, doi:10.1007/s11427-016-9016-6 (2017).
- 8 Cui, S. *et al.* Quercetin inhibits LPS-induced macrophage migration by suppressing the iNOS/FAK/paxillin pathway and modulating the cytoskeleton. *Cell Adh Migr* **13**, 1-12, doi:10.1080/19336918.2018.1486142 (2019).
- 9 Wenzel, J. *et al.* Measurement of TLR-Induced Macrophage Spreading by Automated Image Analysis: Differential Role of Myd88 and MAPK in Early and Late Responses. *Front Physiol* **2**, 71, doi:10.3389/fphys.2011.00071 (2011).
- 10 Jaunin, F., Burns, K., Tschopp, J., Martin, T. E. & Fakan, S. Ultrastructural distribution of the death-domain-containing MyD88 protein in HeLa cells. *Exp Cell Res* **243**, 67-75, doi:10.1006/excr.1998.4131 (1998).
- 11 Weisshaar, N. *et al.* Rgs16 promotes antitumor CD8(+) T cell exhaustion. *Sci Immunol* **7**, eabh1873, doi:10.1126/sciimmunol.abh1873 (2022).

- 12 Narunsky-Haziza, L. *et al.* Pan-cancer analyses reveal cancer-type-specific fungal ecologies and bacteriome interactions. *Cell* **185**, 3789-3806 e3717, doi:10.1016/j.cell.2022.09.005 (2022).
- 13 Kaesler, S. *et al.* Targeting tumor-resident mast cells for effective anti-melanoma immune responses. *JCI Insight* **4**, doi:10.1172/jci.insight.125057 (2019).
- 14 Gaiser, R. A. *et al.* Enrichment of oral microbiota in early cystic precursors to invasive pancreatic cancer. *Gut* **68**, 2186-2194, doi:10.1136/gutjnl-2018-317458 (2019).
- 15 Wu, X. *et al.* Lipopolysaccharide promotes metastasis via acceleration of glycolysis by the nuclear factor-kappaB/snail/hexokinase3 signaling axis in colorectal cancer. *Cancer Metab* **9**, 23, doi:10.1186/s40170-021-00260-x (2021).
- 16 Kalaora, S. *et al.* Identification of bacteria-derived HLA-bound peptides in melanoma. *Nature* **592**, 138-143, doi:10.1038/s41586-021-03368-8 (2021).
- 17 Geller, L. T. *et al.* Potential role of intratumor bacteria in mediating tumor resistance to the chemotherapeutic drug gemcitabine. *Science* **357**, 1156-1160, doi:10.1126/science.aah5043 (2017).
- 18 Nejman, D. *et al.* The human tumor microbiome is composed of tumor type-specific intracellular bacteria. *Science* **368**, 973-980, doi:10.1126/science.aay9189 (2020).
- 19 Mercado-Lubo, R. & McCormick, B. A. A unique subset of Peyer's patches express lysozyme. *Gastroenterology* **138**, 36-39, doi:10.1053/j.gastro.2009.11.033 (2010).
- 20 Chang, Z. Q. *et al.* Endotoxin activates de novo sphingolipid biosynthesis via nuclear factor kappa B-mediated upregulation of Sptlc2. *Prostaglandins Other Lipid Mediat* **94**, 44-52, doi:10.1016/j.prostaglandins.2010.12.003 (2011).
- 21 Hayes, M. P., Wang, J. & Norcross, M. A. Regulation of interleukin-12 expression in human monocytes: selective priming by interferon-gamma of lipopolysaccharide-inducible p35 and p40 genes. *Blood* **86**, 646-650 (1995).
- 22 Grohmann, U. *et al.* Positive regulatory role of IL-12 in macrophages and modulation by IFN-gamma. *J Immunol* **167**, 221-227, doi:10.4049/jimmunol.167.1.221 (2001).
- 23 Dobashi, K. *et al.* Regulation of LPS induced IL-12 production by IFN-gamma and IL-4 through intracellular glutathione status in human alveolar macrophages. *Clin Exp Immunol* **124**, 290-296, doi:10.1046/j.1365-2249.2001.01535.x (2001).

REVIEWER COMMENTS

Reviewer #1 (Remarks to the Author):

MS is revised following my suggestions, I don't have further comments, congratulation for these impressive data set.

Now, I recommend to the editor to accept the MS.

[Editorial note: This reviewer was also asked to comment in place of reviewer 2 who was unavailable to provide a response this round]

I've had an indepth look to the answers of the authors to Reviewer 2. The issues of the reviewer I can follow, nevertheless I'm not an expert in molecular and structural biology, but the authors did additional control experiments, proving whether a potential rescue of Sptlc2-ko macrophages with a mutated MD88 isoform regulates size of these cells. By these approach, the concerns of the reviewer were clarified, the authors are presenting the new data in an additional figure, and do not interpret their data in an inappropriate manner.

Also the other issues of reviewer 2 were appropriately solved, e.g. terminology of 'LPS-induced sepsis', the dynamic and time course of cytokine release, and the discussion, whether the tumor data should be given in a follow-up study,

From my point of view as a 'reviewer-reviewer# I state, that the author answered the questions in an appropriate manner and clarified all of his/her concerns.

Reviewer #3 (Remarks to the Author):

I have no further comments.

Reviewers' Comments:

Reviewer #1 (Remarks to the Author):

MS is revised following my suggestions, I don't have further comments, congratulation for these impressive data set.
Now, I recommend to the editor to accept the MS.

Reply: We appreciate Reviewer #1 for their thoughtful comments that contributed to improving our manuscript, and we highly value the positive feedback.

Editorial note: This reviewer was also asked to comment in place of reviewer 2 who was unavailable to provide a response this round.

I've had an indepth look to the answers of the authors to Reviewer 2. The issues of the reviewer I can follow, nevertheless I'm not an expert in molecular and structural biology, but the authors did additional control experiments, proving whether a potential rescue of Sptlc2-ko macrophages with a mutated MD88 isoform regulates size of these cells. By these approach, the concerns of the reviewer were clarified, the authors are presenting the new data in an additional figure, and do not interpret their data in an inappropriate manner.

Also the other issues of reviewer 2 were appropriately solved, e.g. terminology of 'LPS-induced sepsis', the dynamic and time course of cytokine release, and the discussion, whether the tumor data should be given in a follow-up study,

From my point of view as a 'reviewer-reviewer# I state, that the author answered the questions in an appropriate manner and clarified all of his/her concerns.

Reply: We thank Reviewer #1 for evaluating our answers to Reviewer #2 comments. Also, we are grateful to Reviewer #2 for the comments in the first round.

Reviewer #3 (Remarks to the Author):

I have no further comments.

Reply: Again, we kindly thank Reviewer #3 for the comments in the first round.